# Influence of Rainfall Intensity on the Stability of Unsaturated Soil Slope: Case Study of R523 Road in Thulamela Municipality, Limpopo Province, South Africa

**Fhatuwani Sengani [1,2,\*] and François Mulenga [1]**

1   Department of Electrical and Mining Engineering, Florida Campus, University of South Africa,
    Private Bag X6, Johannesburg 1710, South Africa; Mulenfk@unisa.ac.za
2   Department of Geology and Mining, University of Limpopo, Private Bag X1106, Sovenga 0727, South Africa
\*   Correspondence: fhatuwani.sengani@ul.ac.za or fhatugeorge@gmail.com; Tel.: +27-15-268-3036

**Abstract:** The purpose of this paper was to analyze the impact of extreme rainfall on the recurrence of slope instability using the Thulamela Municipality roads (R523) as a case study. To this end, the historical rainfall data of the area of study were analyzed between 1988 and 2018. The results show that a significant increase in rainfall is usually experienced in the summer months of December and January. Following this, the factor of safety (FoS) of slopes of silt clay, clay, and clay loam soils were estimated using the SLIDE simulator (Numerical software *"Finite Element Method (FEM)"*) under sunny to rainy conditions of the area. A complementary model, FLACSlope (Numerical software *"Finite Difference Method (FDM)"),* was utilized to simulate FoS and pore water pressure in sunny and rainy conditions of the area. Simulation results show that extreme rainfall has the ability to reduce the shear strength and resistance of the soil slope material. This may explain the recurrent landslides noted in the area. Finally, the water pore pressure has been simulated to increase with the increased water table, which generally pushes the soil particles apart and reduces the stress between the particles resulting in soil slope failure. Extreme rainfall alters the phase of the material solid in a manner that may require further research for a better understanding.

**Keywords:** Rainfall intensity; slope instability; numerical modeling; geotechnical; landslides; fluid-solid-structure interaction (FSSI)

## 1. Introduction

Göktepe & Keskin [1] define a landslide as "the downslope movement of soil or rock material under the influence of gravity". Heavy rains, deforestation, and land use are factors that can accelerate landslides. Deforestation for instance can weaken the integrity of the topsoil and reduce slope strength leading to landslides. The most important factor contributing to landslides is arguably the slope angle [1]. The slope angle is directly proportional to the shear stress in the soil or any unconsolidated material. This means that, if the slope angle increases, the shear stress within the material increases, thereby initiating a landslide.

Human factors contribute to landslides through the weakening of slope soil structures, reducing slope strength, and increasing stress placed upon it. A simple example can be through "reducing friction on slopes", "adding weight to slopes", "reducing the resisting mass", and "making the angle of the slope steeper" [2]. All these factors can result in landslides if not adequately monitored [1]. Irrigation, clearing of vegetation, digging, etc. reduces the slope strength [1,3], while slope angle is the most significant component of slope stability analysis [1], and in actual sense the slope angle is

directly proportional to the shear stress in soil or any unconsolidated material. Therefore, if the slope angle increases, the shear stress of the material increases. Some recent studies have made it possible to divide and classify landslides based on several factors, including types and speed of movement, failure mechanisms, etc. Almost all reported high or large landslides are associated with predisposing conditions [4–11]. These include prevailing clay material, the morphological setting of the slope, extreme rainfall events, and human activities such as road construction. Extreme rainfall and snowmelt are however the greatest triggers of the landslides [11]. Nonetheless, understanding of the impact of extreme rainfall on slope stability in unsaturated soil remains to be improved.

Bogaard and Greco [12] have listed rainfall as one of the most common landslides widespread triggering hazards in the world. The previous author [12] supported their arguments by outlining that the hydrology in and around a landslide area is among the factor leading into pore pressure buildup in the soil skeleton, as a result, the shear strength of the soil reduces due "to the buoyancy force exerted by water in a saturated soil and to soil suction in an unsaturated soil". Furthermore, previous authors [12] also documented that the "extraordinary precipitation events trigger most of the landslides, but, at the same time, the vast majority of slopes do not fail" the previous authors [12] suggestion correlates very well with some studies such as those of Sengani and Mulenga [13], Sengani and Zvarivadza [14], and Mutanamba [15]. An encouraging study by Chau et al. [16] suggested that rainfall-triggered landslides tended to occur on dip slopes, instead of the windward slopes, pinpointing geological settings to be the most controlling factor of the mass-wasting processes on hillslope scale than the rainfall condition. This study appears to support or complement one of the suggestions documented by Sengani and Mulenga [13]. Indeed, studies such as those of Bucci et al. [17] have validated that geological features such as faults, which are seismic active. Sophisticated methods such as numerical simulated have been implemented to understand the sliding of clay soil material Kluger et al. [18], and revealed that "the high sensitivity and contributes to an improved understanding of the mechanisms of flow sliding insensitive, altered tephras rich in spheroidal halloysite". In summary, there are several studies [19–21] that document the effect of rainfall and other factors on triggering landslides.

To this end, this paper aims at evaluating the influence of extreme rainfall on the stability of unsaturated soil slope. The R523 road in the Thulamela Municipality, the Limpopo Province, South Africa, has been experiencing a recurrence of slope instability and it served as a case study in this paper. The area was reported in the previous papers (of Sengani and Mulenga, Sengani and Zvarivadza, and Mutanamba) to experience regular slope instability events mostly in summer. However, the mechanism associated with the recurrence of these slope instability events is not well defined.

This paper follows common structures which are followed in the most research paper, in fact besides the introductory section, a brief on the discussion of the locality of the study area is documented, this section is followed by the methodology of the study. The methodology of study commences with a detailed description of visual observations and measurements, the methodology followed in sieve analysis together with Atterberg limits are therefore documented. Further description of the methods used in the study was pointed out by outlining data collection associated with rainfall across the study areas. Finally, detailed description of the numerical simulation conducted in the study is documented. In regard to numerical simulation, the procedure followed in the SLIDEs (FEM) model and the FLACSlope (FDM) model is incorporated in the description of numerical simulation. The above sections have led to the results and discussion section of the paper, this section followed the same order used in the methodology. Nevertheless, the significance of the simulation findings was therefore discussed after the results and discussion. Lastly, the conclusion of the paper is documented.

## 2. Location of the Study Area and Regional Geology Setting

The scope of the study is limited to one national road (R523) located in the Thulamela Municipality, Limpopo province, and South Africa. The Thulamela Municipality is a Category B municipality situated within the Vhembe District in the far north of the Limpopo province as shown in Figure 1.

The Kruger National Park forms its eastern boundary while the southern and south-western side is bordered by the Makhado Municipality.

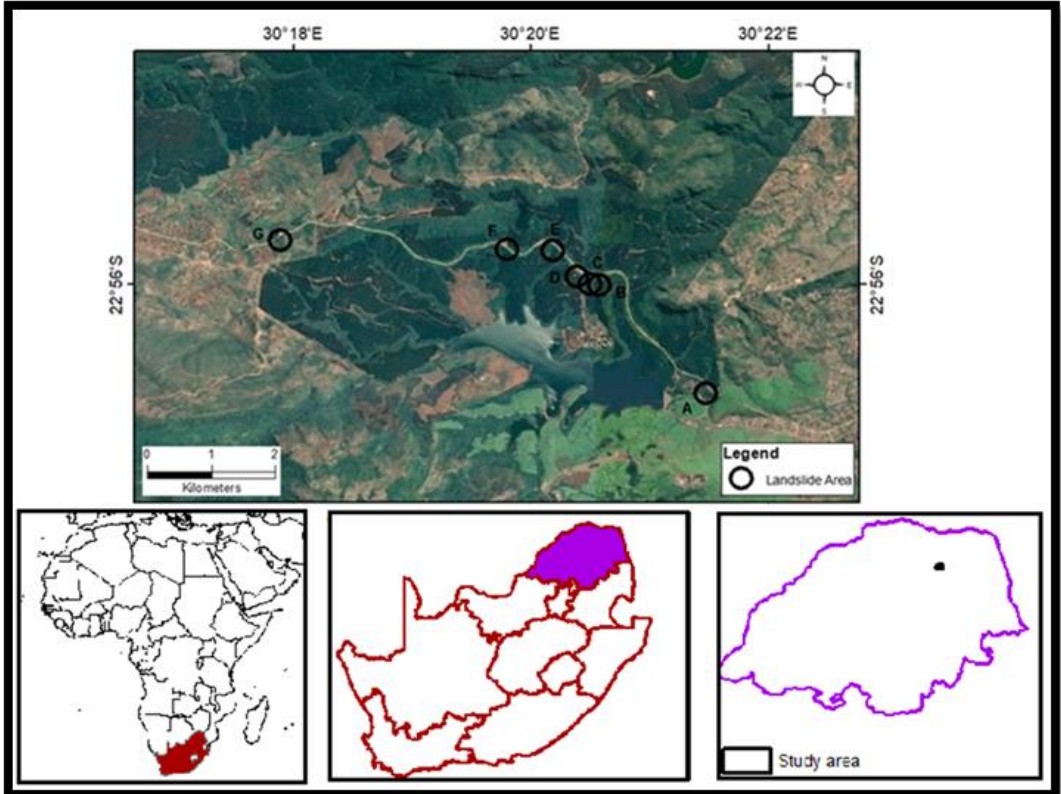

**Figure 1.** Locality map of the study area.

As the smallest of the four municipalities in the district, Thulamela is spread over about 2642 km$^2$. It is, however, the largest in the province in terms of population size [22].

The R523 road identified for this research is situated in the rugged topography of the Southpansberg Group. This geological group is partially buried beneath sedimentary and volcanic rocks of different thicknesses. The group is dominated by large faults and joints thereby creating a discrete and blocky rock mass especially in Thulamela [23–26]. Furthermore, a conjugate fault and joint set has been evidenced in the northwest-southeast and northeast-southwest directions with quartz vein filling the brecciated joints and faults (see Figure 1). It is these geological features that have led to high rockfalls and slope stability problems on the roadside. An attempt is made to understand these phenomena with the help of computer-based simulation tools built around the FEM framework.

This road is the bloodline between villages and surrounding townships. Safety is therefore critical when one considers the fact that they were excavated by the blasting of very sloppy and unstable terrain [14]. The road also passes through several rivers [15]. Intensive summer rainfall and human activities in the study area also make roads susceptible to frequent slope failure.

Several remedial measures have been suggested and implemented but rockfalls and slope instability are still reported at an alarming rate. For example, Mutanamba [15] analyzed the stability of cut-slope along the R523 road between Thathe Vondo and Khalavha. First, geological structures, properties of the soil, and groundwater regime were surveyed. Then, conventional analytical methods were used to classify the type of soil and produce slope stability charts. Finally, two types of failure occurring close to the road were identified from the findings: rotational slump failure and rockfall in the form of toppling. The qualitative findings, however, could not be used to comprehensively describe the underlying mechanism of landslide, occurrence, and deposition. That is what the present research is attempting to explore.

## 3. Material and Method

Detailed methodologies were followed in this paper, these methods include; visual observations and field measurements, rainfall statistical analysis, rainfall departures, and numerical simulation of Thulamela Municipality roads in Limpopo province, South Africa. Detailed methodologies of the study are fully explained below.

### 3.1. Visual Observations and Measurements

Visual observation was extensively used to circumscribe the rockfall problem associated with the selected study areas. Field trips were made to identify the types of soils present in the study areas. The dimensions or extent of the unstable slopes along Thulamela Municipality roads were also measured.

To this end, the width and height of the micro landslides were measured using a geological tape measure. The width and height were measured within the boundaries of the micro landslides. The above-mentioned procedure was followed by collecting soil samples, soil samples were collected random sampling with the emphasis that samples should be collected from the depth of 0.5 m within the affected slopes. In order to simplify the process of soil sampling, a handheld Auger was used to collecting soil samples, with a sample bag of 5 kg. Samples were named from each selected point of the study area so as to avoid the mixing of data, and samples were stored separately. In the meantime, some of the geological features revolving around the study area were recorded by measuring the strike and dip of the feature using campus, other parameters such as the properties of the features including infill, the width of the features were also recorded in a notebook. Geotechnical and geological mapping was also conducted with the focus of identifying dip and dip direction of the wedges developed along with the micro landslides and also measure the slope angle, the orientation of tension cracks.

In terms of actual implementation, three major steps were followed during the geological mapping: planning and preparation; construction of the map in the field; and production of the final map. The planning component of the study was composed of visiting the study area and decided the size of the map as well as designing the traverses that should be followed in the study area using a constructed topographical of the area (see Figure 2). The developed traverses were about 500 m wide and 1.7 km long, and five traverses were developed across the study area. The planning process was limited to the reconnaissance survey so that the starting map of the study area could be drafted. Although this research is not geological in nature, the geology of the study area was crucial to the understanding of the behavior of the material present. After developing reasonable traverses, geological mapping commenced. The traversing process took 2 days as some areas were not accessible while others were dominantly covered with topsoil and fewer outcrops were denoted across the study area.

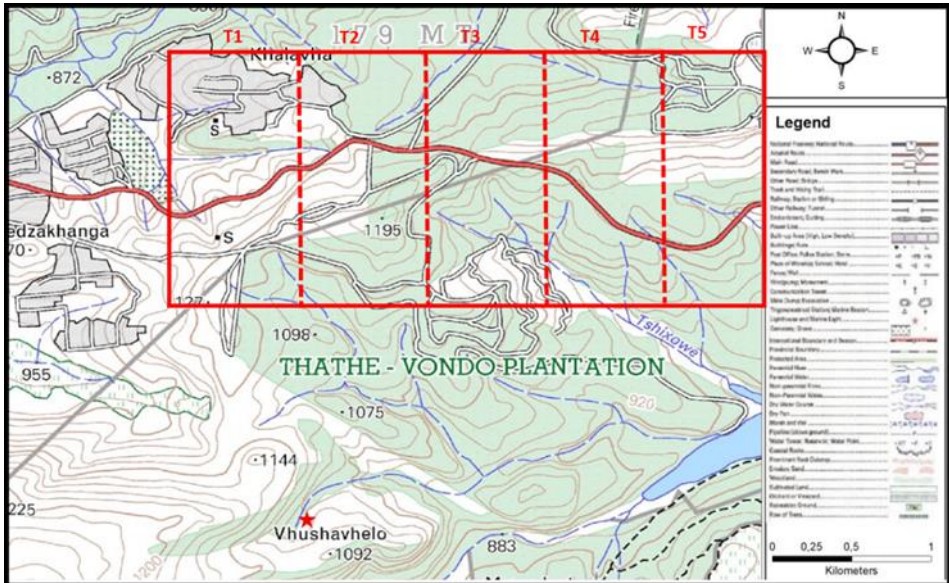

**Figure 2.** Topographical map of the study area with the designed traverses across the area of interest.

### 3.2. Sieve Analysis

Commonly, two distinct classification systems are normally used, these methods include the American Association of State Highway and Transportation Officials (AASHTO) and the Unified Soil Classification System (USCS) [27]. In this paper, the American Association of State Highway and Transportation Officials (AASHTO) was used. Particle size analysis (also known as sieve analysis) was carried out to classify the soil materials, determine their textures, and estimate the size of the grains. The latter is known to greatly influence the behavior of the soil mass; hence, its determination. The following apparatuses were used: a stack of sieves with the pan at the bottom and a top cover, a Sartorius balance accurate to within 0.01 g, a mechanical sieve shaker, a plate, and the Vacutec drying oven. To determine the grain size distribution, the samples were placed in the Vacutec oven and dried overnight. The dried soil was sieved through a stack of sieves placed on the mechanical shaker. Sieves were arranged from top to bottom starting with coarser and ending with a smaller opening. The amount retained on each sieve was collected and weighed. This was later used to calculate the fraction of material passing each sieve as a percentage of the total sample being sieved.

### 3.3. Atterberg's Limits

The objective of testing for Atterberg's limits was to determine the basic index nature of the soil. This gives an idea of the strength and settlement characteristics of the material in-situ. It is also the primary form of classification of fine-grained soils. Atterberg's limits are used to determine the boundaries between the different states of fine-grained soils. The classical states associated with clay are liquid, plastic, semisolid, and solid. However, the laboratory tests were performed to determine only the plastic and liquid limits of fine-grained soil because of the logistical limitation of the laboratory.

The liquid limit (LL) is defined as the water content at which a pat of soil in a standard cup and cut by a groove of standard dimensions' flows together at the base of the groove for a distance of 13 mm (1/2 in.) when subjected to 25 shocks from the cup being dropped 10 mm in a standard liquid limit apparatus operated at a rate of two shocks per second [28]. The liquid limit property of the fine-grained fractions of the samples was tested using a Casagrande liquid limit device, a grooving tool, a water bottle, mixing dishes, a spatula, and the Vacutec drying oven.

The testing procedure began by cleaning up and calibrating the Casagrande equipment. The calibration involved the setting of the drop of the cup to a consistency drop height of 10 mm. The soil was then passed through a #40 sieve, collected, and air-dried. The fraction of soil was

thoroughly mixed with a small amount of distilled water until it turns into a smooth uniform paste. The moist sample was placed on the Casagrande's cup and smoothening to the maximum depth of 8 mm. The groove was cut at the centerline of the sample in the cup. Next, the device was cranked at 2 revolutions per second until the two halves of the soil pat come into clear contact of about 13 mm long at the bottom of the groove. The number of blows (N) that caused the closure was counted and recorded. The sample in the pat was collected, weighted together with the can, labeled, and dried in the oven at 110 °C for 6 h. The weight of the dried sample was also determined. The test was repeated 3 times at different moisture contents; this enabled the production of successively lower numbers of blows to close the groove. The number of blows in each trial was then plotted against the water content. The best-fit straight line through the plotted data points was constructed for which the liquid limit (LL) was determined as the water content corresponding to 25 blows.

In plotting the limited liquid data, some samples do not conform well to the expected straight line. This may be ascribed to the inadequate mass of soil in the brass cup, inadequate rate of blows, or inadequate height of fall. To compensate for this, Equation (1) was used [28]:

$$LL = W_n \left( \frac{N}{25} \right)^{0.121} \tag{1}$$

where *LL* is the liquid limit of the soil, *N* is the number of blows in the liquid limit device, and $W_n$ is the corresponding moisture content.

The plastic limit testing of the soil samples was looked at next. The plastic limit (PL) of soil is defined as the water content at which the soil begins to crumble when rolled into a thread of 3 mm in diameter [28,29].

### 3.4. Influence of Rainfall Intensity on Slope Instability

Since extreme rainfall has been recognized as an important factor in slope stability analysis [8,9], rainfall statistic of the study area was considered for monthly rainfall (mm) data from 1988 to 2018. Rainfall data of the study was therefore collected from the South African weather service (SAWS). The analysis of rainfall patterns helped to determine the months in which the slope failure is likely to occur, and it's rate helped to determine the contribution of water to slope failure in the study area.

### 3.5. Numerical Simulation

Numerical simulation was performed through the used laboratory results, rainfall analysis data, and other field observations inputs. The simulation was sickly focusing on the safety factor and water pore pressure of the defined slope in a case of sunny and rainy conditions. Both finite element methods (FEM) and finite difference methods (FDM) were implemented. In FEM, eight methods were used to quantify the results, those methods are Bishop simplified [30], Janbu Simplified, Janbu corrected [31], Spencer [32], Corp of Engineer number one, and Corp of Engineer Number two [33], Lowe Karafiath [34], and Gle/ Morgenstern Price [35]. All these methods were filtered within the model to produce reliable results of the slope safety factor. To simplify the results of the study, the first analysis was to simulate the safety factor of silt clay soil, clay soil, and lastly clay loam soil. Therefore, a complementary model (FDM) was used to simulate FoS as well as water pore pressure of the slope with a change in the water table. A detailed procedure of the models is outlined below.

### 3.5.1. SLIDES (FEM) Model Procedures

It is however noteworthy to state here that SLIDES was utilized to estimate the FoS values of the slope for several scenarios. The FoS values can be computed following various models (Bishop's, among others). The different outcomes can be compared at once. In terms of model building, the exercise starts with the creation of a new project. Similar to other modeling platforms, the new project required the delimitation of the model limitation in XY coordinates. For that, various X and Y coordinates

defining the region were entered. The ultimate goal of this step was to draw the model of the region. Upon generating the boundaries of the model, the actual initial conditions of the simulation are defined next for the project. Inputs such as the statistics associated with groundwater conditions, the computational methods, and the failure directions are captured.

The correct values of input data and the appropriate selection of procedural approaches were of paramount importance. This is because the exercise determined how realistic the computer modeling of the actual rockmass associated with the study area would be. For a thorough interpretation of the results, several methods (Bishop simplified, Corps of Eng #1&2, GLE, Janbu simplified and corrected, Lowe-K aralias, Ordinary, and Spencer) of analysis were considered. The motivation was to compare various approaches to the problem. However, corresponding results were interpreted based on the performance of each approach.

The other important factor was the groundwater regime set at 9.81 KN per cubic meter. This value was estimated in the laboratory after performing several tests on soil material. Statistics of the yearly rainfalls in the area were also considered in estimating the groundwater regime. It is important to stress that theoretical models underpinning all these computer programs are generally based on simplifying assumptions. These depart from the real behavior of soil and rock masses and diminish the value of output results. To circumvent the limitation, a degree of randomness and unpredictability is supposed to be allocated to the modeling frameworks. In this paper, the exercise was implemented by resorting to the probabilistic analysis built upon the Monte-Carlo sampling method. Indeed, the Monte-Carlo method refers to the allocation of randomness to a phenomenon by means of random number generators.

The next step was the definition of the boundaries of the excavation upon which the roads were constructed. The coordinates entered were based on field surveying and existing maps of the region. The inputs are rendered automatically and diagrammatically as the excavation model. Finally, computational routines are selected and defined in terms of grid spacing of the limit equilibrium analysis of the excavation model. Other relevant input parameters are also incorporated before the model is finally solved. The convergence solution to the problem is exemplified in the results sections with the most likely FoS defined in the inset. Various scenarios were simulated and their output results were recorded for later analysis. Several techniques were used for the purpose and they are presented in the next section.

3.5.2. FLACSlope (FDM) Model Procedures

The model procedure followed in FLACSlope is readily available on the Itasca Consulting Group Inc. website under FLACSlope user guidelines. In short, the procedure followed includes defining the project (project file dialog, slope parameters dialog, model layout), building the model (model layers, properties input in the define material dialog for layers), calculating a factor of safety, and viewing the results. With reference to building the model, the model layers were varied based on the properties of the soil slope modeled, detailed description of the layers' inputs is outlined in Table 1 below. In the meantime, in order to acquire detailed results when calculating FoS, a fine grid model was selected. The water table increased in steps following the rainfall database on the study area. Meanwhile, the material properties of each layer were constant when the water table values increased or varied. Lastly, the FoS, strain, and pore pressure contour of the slope were easily acquired from the model. In addition to Table 1, the volume of the water table has been increased with time up to 2500 $m^3$. In this regard when modeling slope FoS in SLIDEs water table with 1000 $m^3$ was used for a normal condition or sunny condition, while 2500 $m^3$ was used for rainy conditions.

**Table 1.** Material properties of the layers used in the model.

| Parameters | Layers of the Soil Slope | | | |
|---|---|---|---|---|
| | Lower Layer | Upper Soil (Area A) | Upper Soil (Area B) | Upper Soil (Area C) |
| Unsaturated Density (kg/m$^3$) | 1900 | 1600 | 1400 | 1300 |
| Saturated Density (kg/m$^3$) | 2200 | 1900 | 1700 | 1600 |
| Porosity | 0.2 | 0.3 | 0.5 | 0.4 |
| Cohesion (Pa) | 10,000 | 5000 | 8000 | 6000 |
| Friction angle ($^0$) | 30 | 20 | 25 | 27 |
| Soil Particle Density (g/cm$^3$) | 2.8 | 2.65 | 2.63 | 2.66 |

## 4. Results and Discussion

The results of the study have been divided into several sections to unfold the findings and critical discussion of the results.

### 4.1. Initial Results of the Field Observations

The initial walkabout showed that all discovered landslides were located along with the road benches or sidewalls. The ongoing road construction along this mountainous area also appeared to alter the surface drainage patterns, create unstable slopes, and increase runoff (see Figure 3).

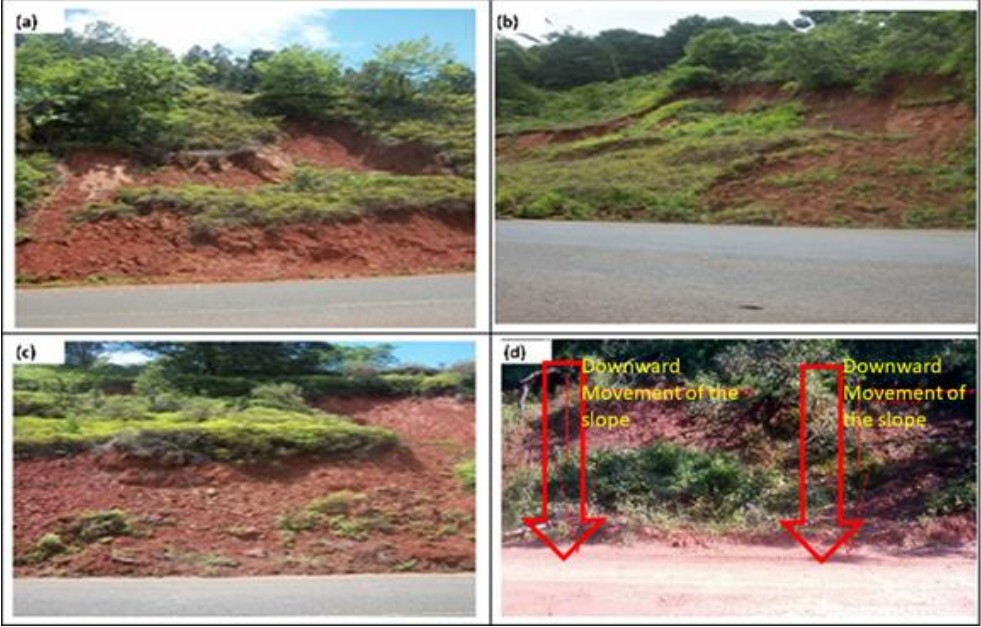

**Figure 3.** Downward movement of material (soil or rocks) along R523 road, (**a**) occurrences of post landslides along the R523 road; (**b**) Tension cracks observed during the reconnaissance survey; (**c**) evidence of an undercut natural slopes become unstable; (**d**) Block of the slope material sliding toward the road.

In simple terms, the first empirical observation suggested that improper road construction could be the reason for the formation of water pools at the foot of the cut-slope. This became a non-negligible driving force for accelerated slope failures. Second, altered drainage patterns led to an increased rate of surface runoff and increased probability of water washing soil down to the road. As a result of this, the surface drainage ditch was blocked while the foot of the slope became waterlogged. The rapid saturation of soil ensued also contributing to the frequent occurrence of landslides. This destabilized the R523 road and the side-cast material downslope; this also made the slope steeper than before. Consequently, undercut natural slopes become unstable. Shear stresses in the soil within the slope increase with increased probability for failure [36].

Third, contrary to local accounts, evidence of previous landslides was noticed also where certain slopes were deemed inactive. This prompted the implementation of mitigating measures to limit further slope failure (landslides). A gabion wall was erected to prevent future landslides. The gabion walls were noted to be effective where two-tiered steps were employed and to fail otherwise. The ineffectiveness of the gabion translated into the collapse of the walls or the wash-away of soil material. In other instances, the failure of the gabion was created to the interruption of traffic due to soil slides blocking the road.

Another factor believed to contribute significantly to slope failure along the study areas is water pressure. Indeed, Duncan and Wright [36] have shown that water naturally fills the cracks developed on the surface of the slope. The water then greatly decreases the shear strength of the soil as pore pressure within the slope increases. Supporting observations also showed that several cracks and a number of short streams were present in most areas with pronounced slope failure.

In summary, the initial field observations enabled us to build an empirical picture of the occurrence of landslides along the R523 road. There is strong evidence that the cross-section profile of the constructed road encourages retention, instead of the flow, of water around the area. And because the majority of landslides occur during the rainy season (i.e., in December and January), the weathering profile of the region is to be further explored. Once this is done in subsequent sections, a correlation between landslide occurrences and rainfall statistics may be established. The benefit of the analysis would be a more realistic simulation of the soil and rock mass.

### 4.2. Geological Description of the Study Area

From the geological mapping, it was established that the study area is dominated with basalts. In contrast, arenite and norite are disturbed along the boundaries of basalt (see Figure 4). Following this, the majority of active unstable slopes were located along with basalts while at the contact zone of basalt and arenite, slopes are more stable. The geological mapping also revealed that post and active slope instability is mostly experienced in red clay soil where basalts are the country-rock. The other interesting observation was that slope failures were mostly composed of soil rather than rock. This reassured the sampling and testing of soil since the collection of core samples was stopped after an equipment failure. From the survey of the target section in Figure 5, a cross-section of the study area was produced. What resulted from this is that layers sitting on top of the country rocks (see Figure 5) were likely to slide when their phase changes from solid to liquid due to rainfalls. One could argue that clay soil on the upper horizon may experience slipping movement along the disturbed boundaries of the study area.

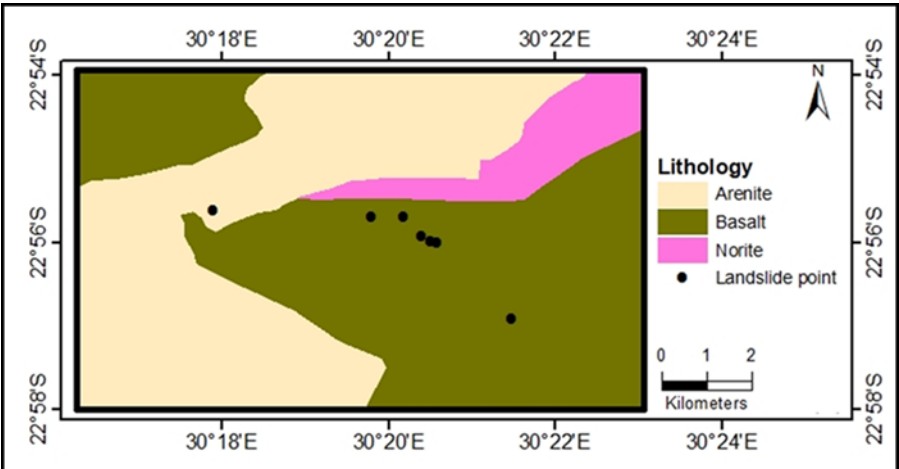

**Figure 4.** Constructed geological through geological mapping.

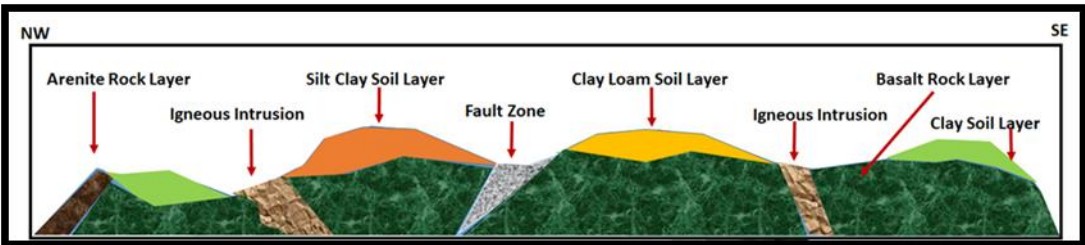

**Figure 5.** Cross-section view of the study area.

A last note on the geological mapping is that road construction in the Thulamela Municipality took place in the mountainous area. Explosives were used for the initial excavation leaving behind fractures and compromised the stability of the rock mass. The fractures generated within the rock mass can readily disintegrate the material when exposed to extreme rainfalls. Large faults and multiple streams cutting across the area were also observed. These features further strain the stability and internal movement of material in the study area. Created tension cracks propagate which in turn weakens the material and leads to gradual movement. Clayey layers capping the country-rock swell when exposed to rainwater and groundwater. The increased internal pressure finally forces rock slopes to collapse. In some cases, the disappearance of vegetation around the slopes is a precursor of a looming landslide.

*4.3. Mechanical Properties of the Soil*

In this section, the mechanical behavior of the solids found in the study area is qualitatively and quantitatively assessed. The following are presented: the particle size distribution, Atterberg's limits, and the classification of the soil.

4.3.1. Particle Size Distribution

Based on laboratory analysis, it was noted that the soil samples tested consisted mostly of clay material. This is because the fraction of material passing the coarsest sieve was high. The particle size distributions also have a small proportion of coarse material or silt soil as shown in Figure 6. Figure 6 further shows that the percent fractions of particles passing through different sieves range from 25% to 100%. It was clearly observed that all samples are finer than the coarsest sieve used for laboratory analysis, i.e., 1 mm. The fraction of soil passing through various sieves gradually also reduces with particle size. All the above suggest that the sample contained a fair fraction of pure and finely-grained clay soil.

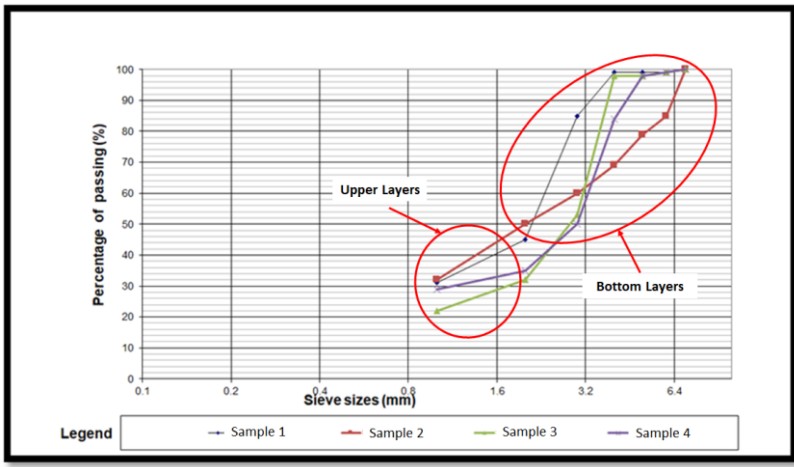

**Figure 6.** Gradation curve of the collected soil sample. The focus of this paper is on the upper layers. It is crucial to indicate that the gradation curves represent the mixture of upper/top layer to the bottom layer of the soil mass.

In summary, all samples were mostly dominated with fine clay material and a small proportion of silt size material. This type of soil is known to be sensitive and prone to experience failure when exposed to extreme rainfall.

### 4.3.2. Atterberg's Limits

Atterberg's limits are a set of index tests performed on finely grained soils to determine their relative activity to moisture. In this section, liquid limits, plastic limits, and plastic index are presented. It is important to state that soil samples collected across the study area were found to have similar Atterbeg's limits. Indeed, the soil collected along the R523 roads were generally found to have 70% of the liquid limit on average (see Table 2). This result confirms that the soil is a very fine and clayey material with the ability to absorb large amounts of water within its pore spaces. Consequently, the accumulated water takes time to pass through or evaporate for this type of soil. Such soil also displays slipping and swelling characteristics especially in the presence of water. Conversely, this type of soil easily cracks when it dries up. Note that, although the fourth sample appears to present a less liquid limit, in reality, it falls under the same category. The causative factor for this is the slightly higher amount of silt particles present in the sample.

**Table 2.** Summary of Atterberg's limits of soil samples.

| Samples | Liquid Limit (%) | Plastic Limit (%) | Plasticity Index (%) | Description |
|---|---|---|---|---|
| Sample 01 | 71 | 37 | 34 | High plasticity |
| Sample 02 | 66 | 34 | 32 | High plasticity |
| Sample 03 | 70 | 37 | 34 | High plasticity |
| Sample 04 | 56 | 26 | 30 | High plasticity |

Liquid limit alone cannot be used to draw a solid conclusion on the mechanical behavior of soil. It should be complemented with the analysis of the plastic limit and plastic index.

The plastic limit can be regarded as an indicator of the water content at which the soil begins to crumble. Based on the results in Table 2, it can be seen that all samples have a plastic limit ranging between 26% and 37%. The soil material of the entire study area can therefore be categorized as of high plasticity in accordance with Table 2. Indeed, the thread could be rolled out down to 3.2 mm at any moisture possible during the experiment. The soil has the ability to contain water at a very low percolation rate; it becomes plastic even after a short rainfall and does not hold well at a high slope angle. Plastic limit results also correlate well with liquid limit ones.

Talking about the plastic index, it can be said that it indicates the range of moisture content over which the soil is in a plastic state. Based on the work of Das [28], the plastic index is calculated as the numerical difference between the liquid limit and the plastic limit of the soil. Liquid and plastic limits are both dependent on the amount and type of clay in the soil, but the plasticity index is generally only dependent on the amount of clay present.

The results in Table 1 show that the plasticity index of the material is at an average of 32% which is a classified as high plasticity index. This was confirmed by plotting the results in Table 1 on the plasticity index chart, as shown in Figure 7.

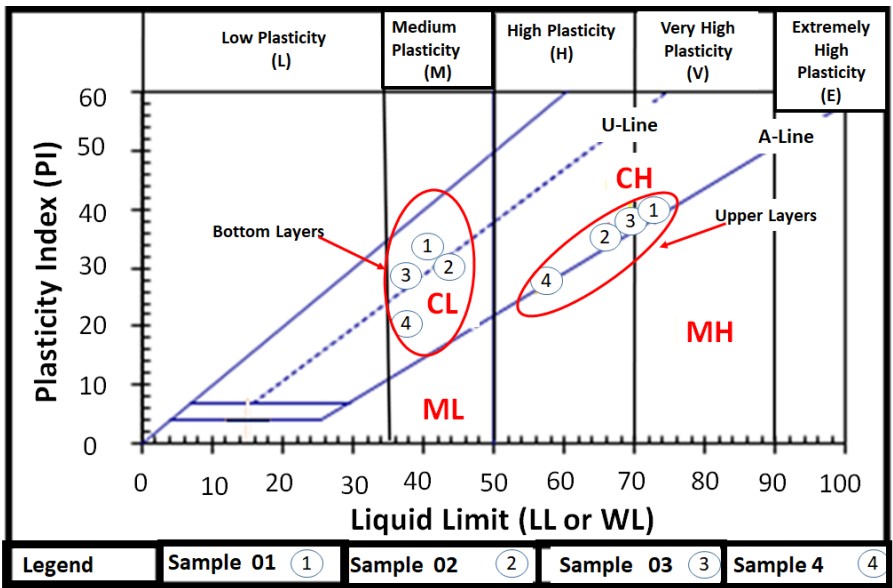

**Figure 7.** Plasticity index charts of collected soil samples. The focus is on the upper layer zone of the soil mass.

It is important to note that the Atterberg's limits presented in this section mostly give a qualitative behavior of the soil subjected to different conditions.

### 4.3.3. Soil Texture and Soil Types

Based on observations, the material found in the study area was classified. It consisted of soil significantly rich in silt clay, clay loam, and clay. In an attempt to be less subjective, the soil was also classified based on soil classification charts in Figure 8. It appears that the clayey soil within the study area is indeed prone to landslides. The soil also may become oversaturated under heavy rains, and may thereafter swell and slip easily. Cracks from drying and subsequent shrinkage are also to be expected.

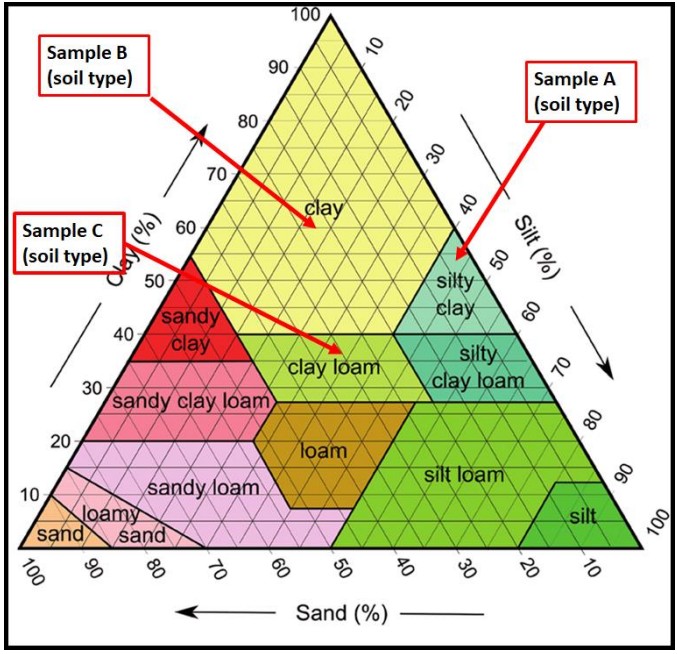

**Figure 8.** Soil texture chart of the study area.

### 4.3.4. Mechanical Properties of Soil

Input parameters for use in numerical simulations necessitated the soil samples to be further analyzed. To this end, additional standard tests were done which resulted in the properties listed in Table 3. By and large, their values for three sites considered were closely comparable. Differences attributable to site locations were noted for the cohesion, the shear strength, and the undrained compressive strength. A detailed procedure on how to obtain each parameter can be found in previous studies, such as those of Fang and Daniels [37].

**Table 3.** Mechanical properties of soils in areas A, B, and C.

| Parameters | Area A | Area B | Area C |
|---|---|---|---|
| Density (kg/m$^3$) | 1900 | 1600 | 1700 |
| Unit weight (kN/m$^3$) | 18 | 18 | 18 |
| Poisson's ratio ($v$) | 0.39 | 0.39 | 0.39 |
| Young's modulus E (MPa) | 3 | 3 | 3 |
| Undrained compressive strength (kPa) | 66 | 55 | 45 |
| Shear strength (kPa) | 250 | 110 | 90 |
| Cohesive strength (kPa) | 96 | 188 | 95 |
| The angle of internal friction | 20° | 20° | 20° |
| Compressibility | 0.17 | 0.15 | 0.17 |
| Soil sensitivity | 1 | 1 | 1 |

One noteworthy observation from Table 3 is that the types of soil within the study area can be classified as sensitive. This is corroborated by the findings reported in Section 4.3.2 whereby the soil contains large proportions of clay. Kaolinite and hematite were also noted to be the main forms of clay minerals in the study area. Such types of clay soil are usually associated with basalts and other igneous rocks [38,39]. Initial laboratory analysis further confirms that the geological mapping results in Section 4.2 make sense. Although the study area is clayey, silt and loam soils are also present. These have generally contributed to undrained compressive strength, cohesive strength, shear strength, and compressibility of the soil.

In conclusion, the soil material is not favorable to geotechnical work. This is because of its poor consistency and likelihood to fail when exposed to extensive rainfall or seismic movement. The qualitative description of the soil provides an understanding of the numerous slope failures observed along the sidewalls of the roads. It also gives insight into the unsuccessful mitigating measures against slope failures in the study area. The suggestion is that the soil properties and the morphological terrain of the study area could be the primary controlling factors of the recurrence of the slope instability.

Finally, in terms of landslide patterns, two types were noted: active slope instability and post slope instability. And although detailed slope stability analysis is discussed in the coming section, the initial findings reported here have enabled the identification of soil properties as a great contributor to slope instability.

### 4.4. Rainfall in the Thulamela Municipality

The analysis of rainfall patterns was intended to determine the months during which slope failure is more likely to occur. The monthly rainfall over a period spanning from 1988 and 2018 in the Thohoyandou area varied between 20 mm and 1100 mm. As reported in Figure 9, most of the annual rainfall is received in the summer months, that is, between November and February. The rainfall departure of the study area was also constructed, as shown in Figure 10. This was done to verify whether extreme rainfall had occurred within specific months.

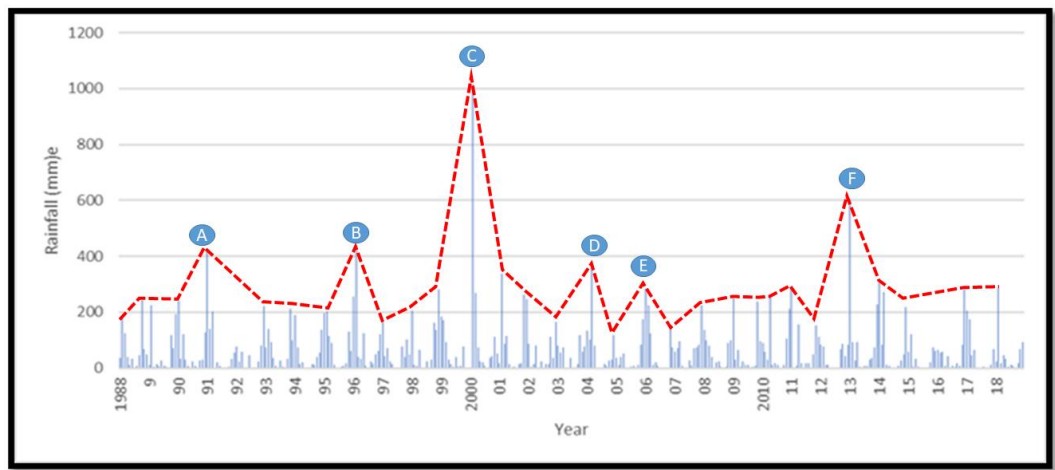

**Figure 9.** Monthly rainfall statistic for Thohoyandou area from the year 1988 to 2018.

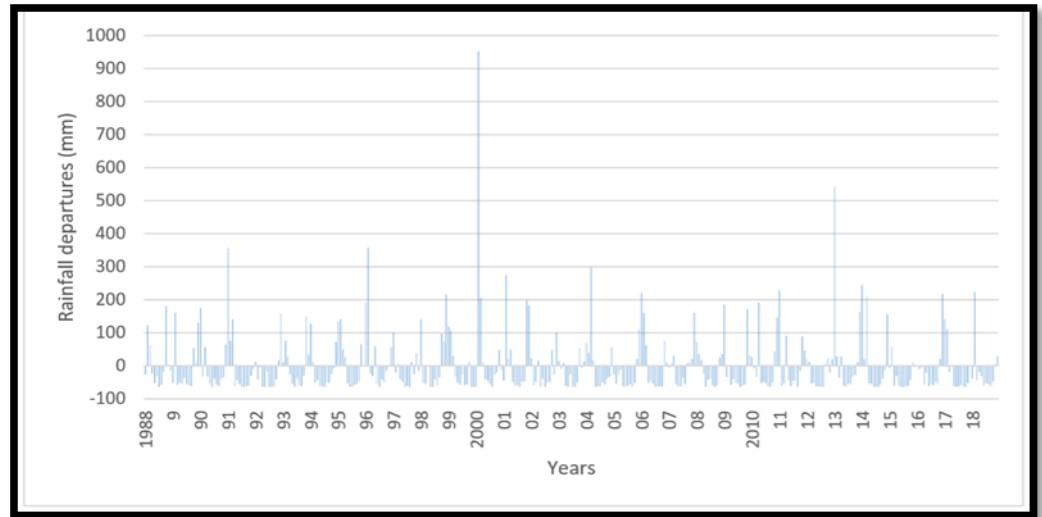

**Figure 10.** Rainfall departures of Thohoyandou area from 1988 to 2018.

Early rains are usually absorbed into the soil through the pores that get filled with water very quickly. Therefore, extreme rainfall in the study area can easily trigger landslides. Although some of the road sidewalls have been supported using gabions, these usually wash away during mass movement. Note that mass movement is commonly triggered by extreme rainfalls.

It can be argued that heavy rainfalls create significant positive pore pressure; this then reduces the frictional shear resistance of the material and results in slope instability across the study area. The underground water also receives a constant supply of rain to form periodical streams meandering across a steep slope. This further contributes to widespread slope instability in the area. The rainfall departures were used as the volume of the water table. However, the volume used was in the form of accumulated volume.

In the next section, numerical modeling was resorted to with the view to explore the impact of extreme rainfall on the rock slope instability. To align the analysis with the study area, the soil properties obtained within the study area and reported in Tables 2 and 3 were used. The study area was assumed to experience drained conditions, but undrained conditions were also simulated to obtain a full view of the impact of rainfall within the simulation. The results of the simulations are discussed below.

### 4.5. Simulation of the Effects of Rainfall Intensity on Slope Stability

This section is devoted to the numerical simulation of the contribution of rainfall to the stability of soils of different compositions. The soils under consideration are characteristic of the locality along the R523 roads in the Thulamela Municipality. Simulations entail the estimation of FoS values of the different slope compositions subjected to drained and undrained conditions. The effects of rainfall intensity are tested on silt clay soil, clay soil, and loam soil slopes.

#### 4.5.1. Simulation Case of a Clay Soil

From Section 4.4, it was conjectured that extreme rainfall has been influencing the recurrence of slope instability within the study area. To test this, numerical simulations were used to estimate the safety factors of various soil slopes of the area under sunny and rainy conditions. Simulation models included the following: finite element method "SLIDEs" (Bishop's simplified, Janbu's Simplified, Janbu's corrected, Spencer's, Corp of Engineers' Number One and Number Two, Lower Karafiath and Gle/Morgenstern Price) and finite difference method "FLAC Slope". The summary of the results is shown in Table 4 for both FoS simulated using FEM and FDM. However, a detailed description of each simulation is denoted in the subsections below. In Table 4, the FoS of the slope is denoted using different methods subjected to change in volume of the water table. In this regard, rainfall departure ranging from 1000 m$^3$ to a maximum of 2500 m$^3$ were implemented or used the input parameters for the water table. However, these values were used as accumulated volume. Furthermore, the simulation using an initial volume of rainfall which is considered to be in normal condition has documented in the table meanwhile the actual simulation are not listed to avoid a large number of images, readers are referred to Sengani and Mulenga [14], for the simulation in normal conditions. It is, therefore, crucial to indicate that the initial and the final volume of rainfall water was used in the SLIDEs model meanwhile the variety of rainfall volume was used in FLACSlope.

Furthermore, the simulation using FDM was intended to outline the behavior of the slope with an increase in the volume of water, however, the model (FLACSlope) has the ability to provide a simulation of Safety Factor and slope displacement. This model was there used to critically analyze the influence of rainfall intensity on the stability of the slope. The alternative method was to implement eight limit equilibrium method using the SLIDEs model in simulating the Safety factor of the slope when the slope was exposed to rainfall with a maximum of 2500 m$^3$ of water or rainfall. As already indicated above, a summary of the results is denoted in Table 4, and a more detailed discussion is outlined in the respective subsections.

Simulations of the FoS Using SLIDE Model under Sunny Conditions in Clay Soil Slope

With reference to Table 4, FoSs values of the slope with clay soil were simulated using nine methods as indicated in Table 4. The simulations have revealed that the slope is unstable or prone to fail, such failure is believed to be influenced by the properties of the material, as well as the properties of the slope. In supporting the simulation, field observations (see Figure 3) whereby the extent of slope failure was very wide in clay soils than in other soils, indeed the properties of soil material could have been played a major role. In summary, the simulation results have some correlation with empirical analysis and observations. This provides a certain degree of reliability in the simulation model produced in this study. Furthermore, simulation in rainy conditions with the increase in rainfall volume or accumulated water table was based on rainfall departure.

**Table 4.** Simulated FoS of the slope using different methods.

| FEM (SLIDE 2D) | FoS in Sunny Conditions | | | FoS in Rainy Condition | | | FDM | FoS | | |
|---|---|---|---|---|---|---|---|---|---|---|
| Slope Composition | Clay Slope | Silt Clay | Loam Clay | Clay Slope | Silt Clay | Loam Clay | Water Level | Clay Slope | Silt Clay | Loam Clay |
| Bishop's simplified | 0.659 | 1.536 | 1.469 | 0.229 | 0.284 | 0.257 | 1000 m$^3$ | 1.42 | 1.94 | 1.99 |
| Janbu's Simplified | 0.612 | 1.424 | 1.364 | 0.203 | 0.264 | 0.228 | | | | |
| Janbu's corrected | 0.643 | 1.497 | 1.434 | 0.221 | 0.284 | 0.247 | 1500 m$^3$ | 1.37 | 1.76 | 1.72 |
| Spencer | 0.760 | 1.675 | 1.610 | 0.231 | 0.306 | 0.257 | | | | |
| Corp of Engineers' Number One | 0.833 | 1.708 | 1.640 | 0.256 | 0.350 | 0.303 | 2000 m$^3$ | 1.17 | 1.54 | 1.51 |
| Corp of Engineers' Number Two | 0.894 | 1.717 | 1.654 | 0.263 | 0.340 | 0.327 | | | | |
| Lower Karafiath | 0.731 | 1.636 | 1.572 | 0.246 | 0.297 | 0.327 | 2500 m$^3$ | 0.79 | 1.20 | 1.25 |
| Gle/ Morgenstern Price | 0.721 | 1.636 | 1.540 | 0.231 | 0.296 | 0.257 | | | | |

Simulations of the FoS Using SLIDE Model under Rainy Conditions in Clay Soil Slope

Simulation scenarios similar in conditions to those in the previous section were set up. The difference is that this time, the clay soil typical to the Thulamela area was tested. The outcome of the simulation work is summarised in Figures 11–14. For all computational methods used, it can be seen that FoS values are all below 0.3 and range between 0.203 and 0.263. This means that the soil type is also prone to failure even under moderate raining conditions. In-situ observations support the simulation outcomes with evidence of multiple tension cracks around the area. In simple terms, clay soil slopes in the area do not require a lot of water pressure in order to collapse.

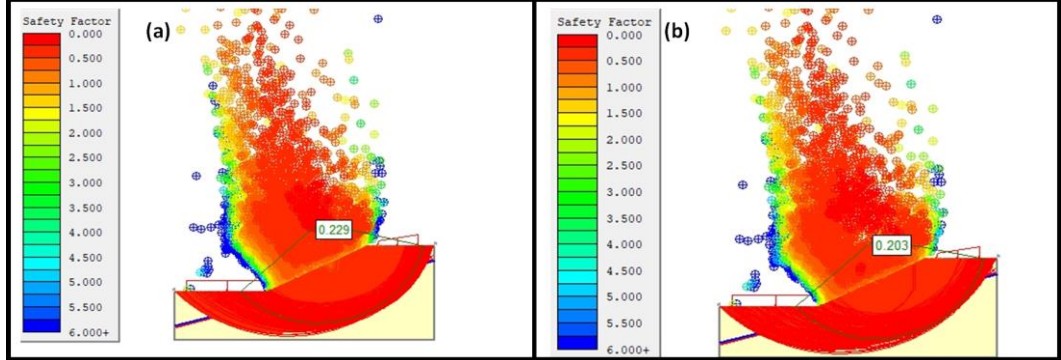

**Figure 11.** Safety factor simulated for rainy conditions in clay soil slope using (**a**) Bishop's simplified method and (**b**) Janbu's simplified method. The FoS simulated using the two methods suggested unstable slope, which could have been affected by the increase in water table volume.

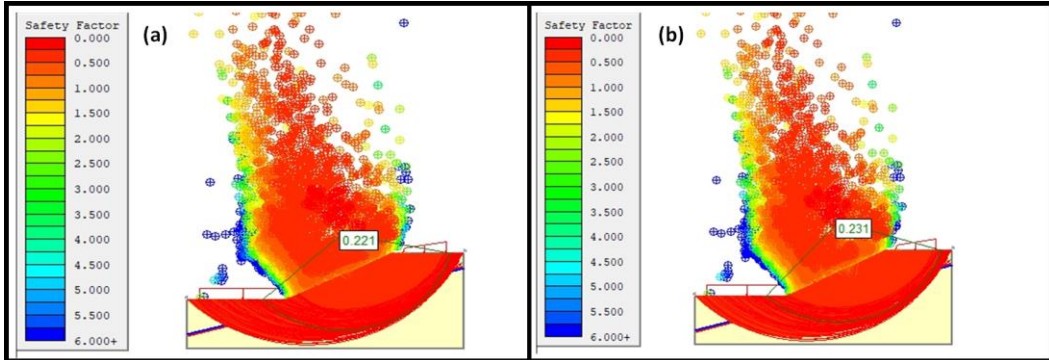

**Figure 12.** Safety factor simulated for rainy conditions in clay soil slope using (**a**) Spencer's method and (**b**) Corp of Engineers' Number One. The two methods also agree with the previous two techniques (see Figure 11a,b) used to simulate the FoS.

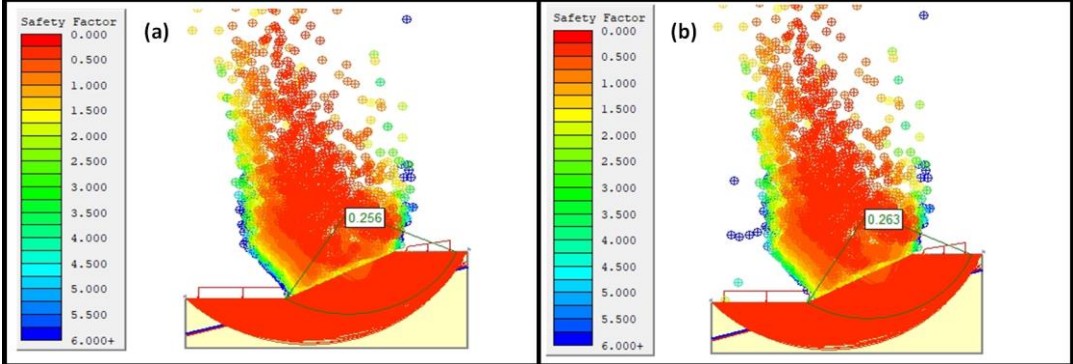

**Figure 13.** Safety factor simulated for rainy conditions in clay soil slope using (**a**) Corp of Engineers' Number Two method and (**b**) the Lower Karafiath method.

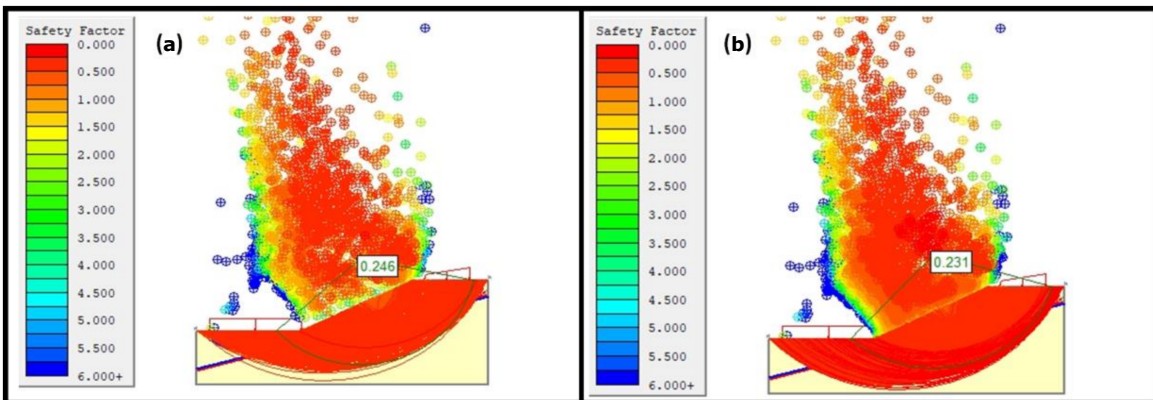

**Figure 14.** Safety factor simulated for rainy conditions in clay soil slope using (**a**) Lower Karafiath method and (**b**) the Gle/Morgenstern Price method.

Simulations of the FoS Using FLACSlope (FDM) Model under Sunny to Rainy Conditions in Clay Soil Slope

Further simulations of FoS have been conducted using the Finite Difference Method (FLACslope), with reference to Figures 15–18. The simulation was performed in both sunny and rainy conditions, nevertheless, sunny conditions were simulated then followed by rainy conditions in which an increase in water table was implemented to identify the behavior of the slope with change in rainfall or extreme rainfall. The results of the simulations have revealed that during sunny conditions the simulated slope could be considered unstable with the FoS of about 1.42 (see Figure 15 and Table 5). It is crucial to indicate that this FoS gives the impression that the slope is not stable however the slope cannot be categorized as a collapsed slope yet. Nevertheless, the slope is actually closer to the border of stability. Therefore, it could be assumed that the slope is less stable in this condition.

This brings us to discuss the results of the simulation in rainy conditions with the increase in the water table. With the reference to Figures 16–18, it was discovered that as the water table increases with time the FoS rapidly decreases (FoS range from 1.37 to 0.79) leading the slope to collapse. The results of the simulation correlate very well with the results simulated by finite element methods in rainy conditions (see Figures 16–18). Nevertheless, it has been revealed that the slope does not immediately collapse, but this failure of the slope seem to occur with time when the material is changing in phases (solid-like to liquid-like then solid-like) due to accumulation of water or increase in the water table. Arguably, the situation was addressed by several numerical techniques, meanwhile, the technique seems to fully agree to each other, therefore this gives an impression that the slope failure within Area A could have been highly influencing by the extreme rainfall. However, extreme rainfall is not the only factor, there are other factors in which the scope of the study is not intended to look into them this time.

Finally, Lazzari and Piccarreta [10] and Lazzari et.al. [11] argued that hydrometric streams usually play a major role in triggering landslides in most sensitive clay soil. Although in some places, the increase in rainfall may not necessarily mean a landslide will occur. The understanding is that the soil phase changes with time but depends on the amount of rainfall filling the pore spaces. The latter is also controlled by the existence of water streams cutting across certain lithologies in the area. The influence of pore pressure on the stability of the slope is discussed in the last sections of results and discussion.

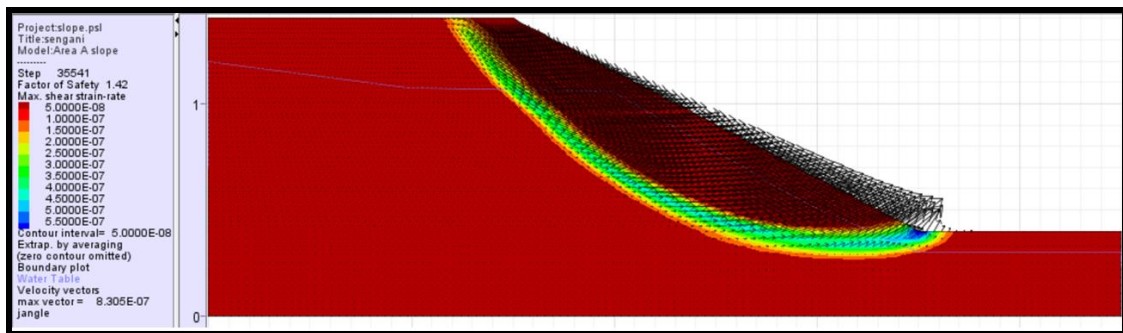

**Figure 15.** Simulated FoS of the clay soil slope using FLACSlope in sunny conditions with a water table with 1000 m$^3$ of water. Note that maximum shear strain simulated ranges between $5.0 \times 10^{-8}$ to $8.30 \times 10^{-7}$.

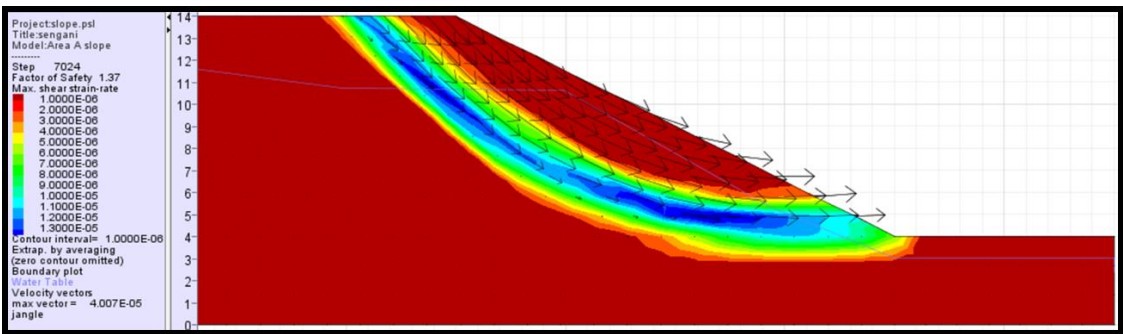

**Figure 16.** Simulated FoS of the clay soil slope using FLACSlope in rainy conditions with water table with 1500 m$^3$ of water. Note that maximum shear strain simulated ranges between $1.0 \times 10^{-8}$ to $4.007 \times 10^{-5}$.

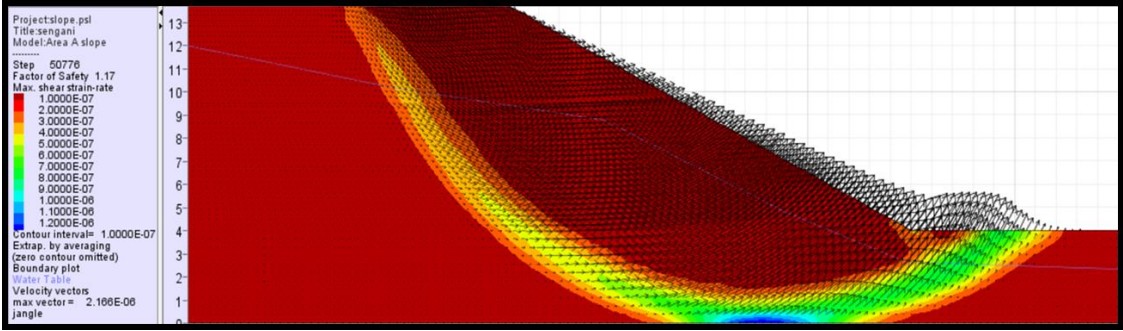

**Figure 17.** Simulated FoS of the clay soil slope using FLACSlope in rainy conditions with water table with 2000 m$^3$ of water. Note that maximum shear strain simulated ranges between $1.0 \times 10^{-7}$ to $2.166 \times 10^{-6}$.

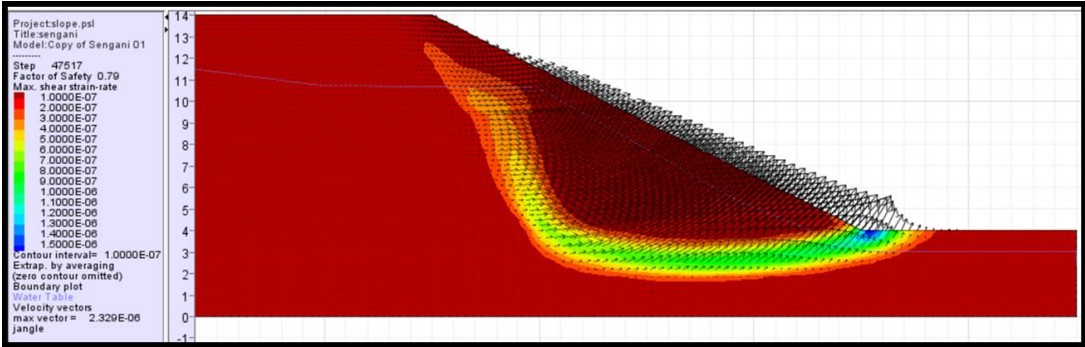

**Figure 18.** Simulated FoS of the clay soil slope using FLACSlope in rainy conditions with water table with 2500 m$^3$ of water. Note that maximum shear strain simulated ranges between $1.0 \times 10^{-7}$ to $2.329 \times 10^{-6}$.

**Table 5.** General results of FLACSlope simulations in clay soil slope.

| Factor of Safety (FoS) | Volume of Rainfall (m³) | Slope Max Shear Strain (Displacement in Shear) | |
| --- | --- | --- | --- |
| | | Minimum Shear Strain | Maximum Shear Strain |
| 1.42 | 1000 | $5.0 \times 10^{-8}$ | $8.30 \times 10^{-7}$ |
| 1.37 | 1500 | $1.0 \times 10^{-8}$ | $4.007 \times 10^{-5}$ |
| 1.17 | 2000 | $1.0 \times 10^{-7}$ | $2.166 \times 10^{-6}$ |
| 0.79 | 2500 | $1.0 \times 10^{-7}$ | $2.329 \times 10^{-6}$ |

### 4.5.2. Simulation Case of a Silt Clay Soil

The second simulations were undertaken to verify the performance of the typical clay loam soil subjected to different conditions (sunny and rainy). Similarly, from the previous discussion when simulating using SLIEDEs, in the sunny conditions water table with a volume of 1000 m³ was used followed by a rainy condition with a water table with a volume of 2500 m³. On the other hand, the simulation using FLACSlope has considered the entire accumulated volume of the water table from 1000 to 2500 m³. The results of the simulation are discussed in the subsections below.

#### Simulations of the FoS Using SLIDE Model under Rainy Conditions in Silt Clay Soil Slope

The results of the simulation (see Table 4) denoted the FoS values range from 1.424 to 1.717, the results indicate that the simulated slope is stable. Similarly, with the previous discussion on clay soil, a correlation between visual observation and the simulated results were performed, it noted that there is some correlation between the two, however it was also noted that although the model estimated stable slope, the stability of the slope is usually compromised when subjected to heavy rainfall which increases the volume of the water table. However, the extent of slope failure is moderate compared to pure clay soils. Detailed simulation of the slope when subjected to an increase in the water table of 2500 m³ is documented below.

#### Simulations of the FoS Using SLIDE Model under Rainy Conditions in Silt Clay Soil Slope

Figures 19–22 give a summary of the simulated scenarios as well as the associated FoS for the silt clay soil. It can be seen that FoS values range from 0.264 to 0.350. So, regardless of the computational methods used, the slope in this silt clay soil is deemed unstable. This confirms that the selected type of slope is highly prone to landslides in the area. Furthermore, it was also noted that Spencer's and Corp of Engineers' methods produced closely comparable FoS values. However, by and large, FoS estimates are similar to within ±0.2 units (i.e., 0.264 < FoS < 0.350).

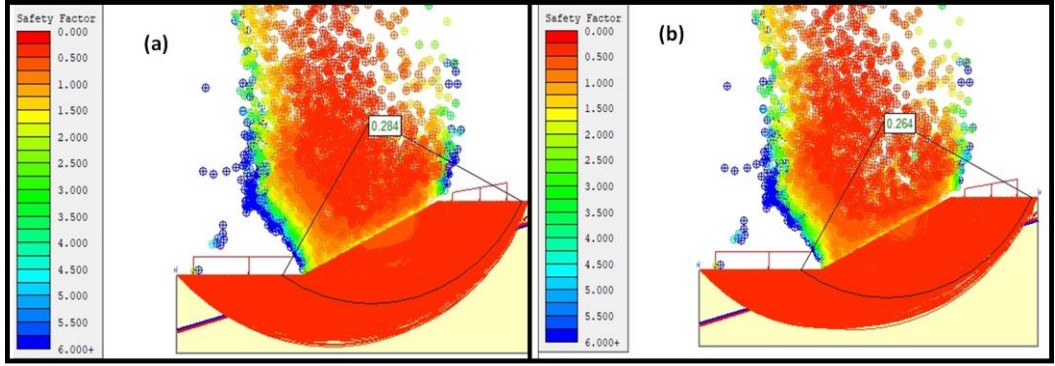

**Figure 19.** Safety factor simulated for rainy conditions in silt clay soil slope using (**a**) Bishop's simplified method and (**b**) Janbu's simplified method.

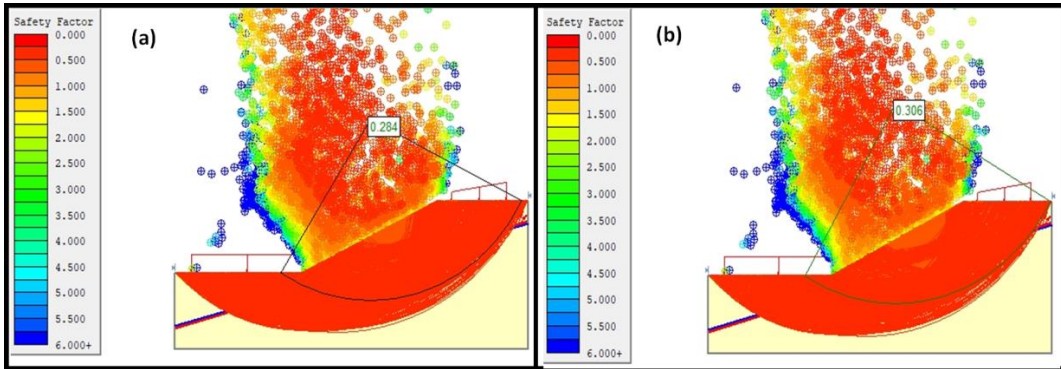

**Figure 20.** Safety factor simulated for rainy conditions in silt clay soil slope using (**a**) Spencer's method and (**b**) Corp of Engineers' Number One method.

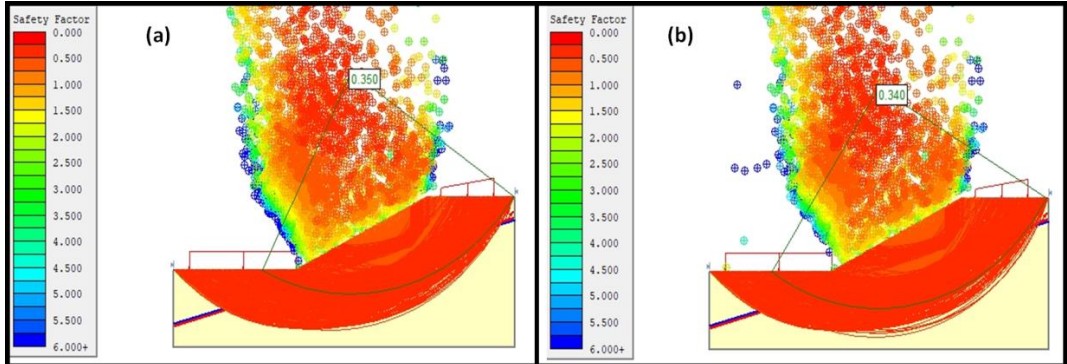

**Figure 21.** Safety factor simulated for rainy conditions in silt clay soil slope using (**a**) Corp of Engineers' Number Two method and (**b**) Lower Karafiath method.

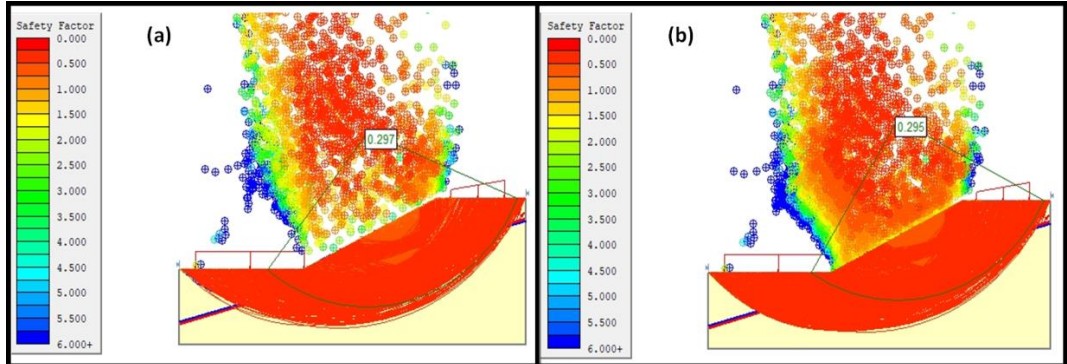

**Figure 22.** Safety factor simulated for rainy conditions in silt clay soil slope using (**a**) Lower Karafiath method and (**b**) Gle/Morgenstern Price method.

Simulations of the FoS Using FLACSlope (FDM) Model under Sunny to Rainy Conditions in Silt Clay Soil Slope

Figures 23–26 and Table 6 gives a summary of the simulated scenarios as well as the associated FoS for the silt clay soil from sunny to extreme rainy conditions. It can be seen that FoS values range from 1.94 to 1.26. The simulation has shown that the slope is stable in sunny conditions, however, the stability of the slope reduced gradually as the water table increases due to extreme rainfall. This result give the impression that, regardless of the computational methods used, the slope in this silt clay soil is deemed unstable when exposed to extreme rainfall. Further compromise with the FEM method has revealed that the slopes within the selected areas are prone to failure especially when the rainfall occurs continuously.

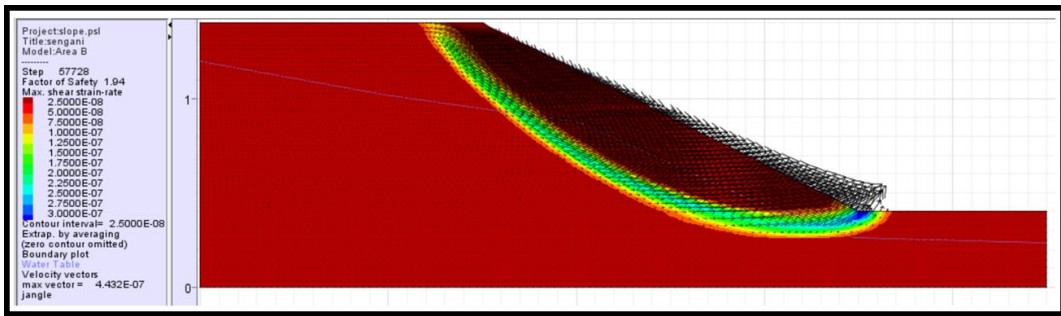

**Figure 23.** Simulated FoS of the silt clay soil slope using FLACSlope in sunny to rainy conditions with water table with 1000 m$^3$ volume of water. Note that maximum shear strain simulated ranges between $2.500 \times 10^{-8}$ to $4.432 \times 10^{-7}$.

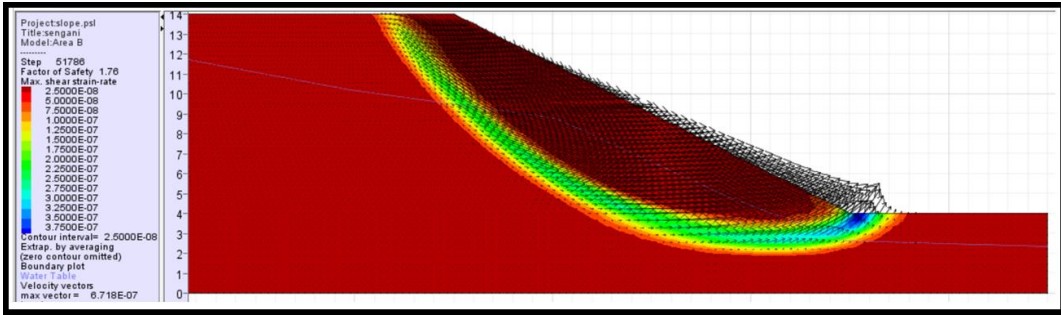

**Figure 24.** Simulated FoS of the silt clay soil slope using FLACSlope in rainy conditions with a water table with 1500 m$^3$ volume of water. Note that maximum shear strain simulated ranges between $2.500 \times 10^{-8}$ to $6.718 \times 10^{-7}$.

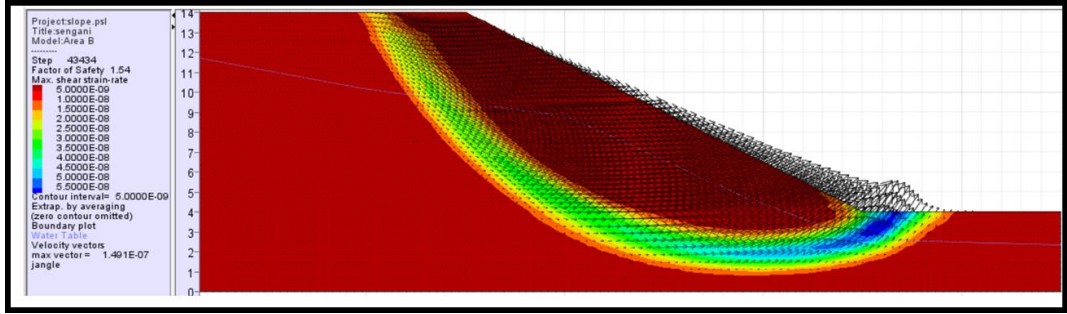

**Figure 25.** Simulated FoS of the silt clay soil slope using FLACSlope in rainy conditions with a water table with 2000 m$^3$ volume of water. Note that maximum shear strain simulated ranges between $5.000 \times 10^{-9}$ to $1.491 \times 10^{-7}$.

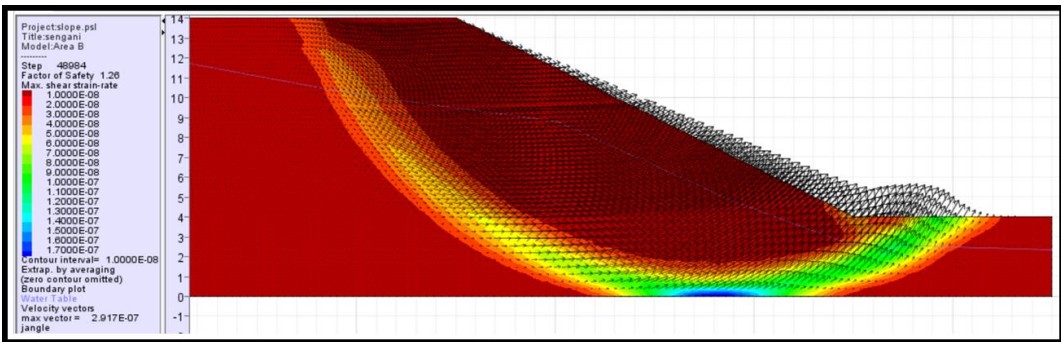

**Figure 26.** Simulated FoS of the silt clay soil slope using FLACSlope in rainy conditions with a water table with 2500 m$^3$ volume of water. Note that maximum shear strain simulated ranges between $1.0 \times 10^{-7}$ to $2.917 \times 10^{-7}$.

**Table 6.** General results of FLACSlope simulations in clay soil slope.

| Factor of Safety (FoS) | Volume of Rainfall (m$^3$) | Slope Max Shear Strain (Displacement in Shear) | |
| --- | --- | --- | --- |
| | | Minimum Shear Strain | Maximum Shear Strain |
| 1.94 | 1000 | $2.500 \times 10^{-8}$ | $4.432 \times 10^{-7}$ |
| 1.76 | 1500 | $2.500 \times 10^{-8}$ | $6.718 \times 10^{-7}$ |
| 1.54 | 2000 | $5.000 \times 10^{-9}$ | $1.491 \times 10^{-7}$ |
| 1.26 | 2500 | $1.0 \times 10^{-7}$ | $2.917 \times 10^{-7}$ |

The simulation results in Figures 23–26 show a rapid drop in FoS. This is indicative of the contribution of rainfall to the FoS which leads to a less stable slope. To put it another way, landslides are more likely to occur in this material when extremely wet. In summary, historical rainfall data and visual observations corroborate the simulation results. It is posited that the soil phase quickly changes during heavy rains. This turns the soil into a liquid-like material. At this stage, the shear strength rapidly drops and weakens the bond between particles. Finally, deformation ensues in the form of a landslide.

### 4.5.3. Simulation Case of a Clay Loam Soil

The last simulations were undertaken to verify the performance of the typical clay loam soil. The results of the simulation have also followed a similar procedure to the previous section. A detailed description and discussion of the results are documented below.

#### Simulations of the FoS Using SLIDE Model under Sunny Conditions in Clay Loam Soil Slope

Last, the simulation was performed to verify the stability of the slope in sunny conditions within clay loam soil. Indeed, the results of the simulation (see Table 4) correlated very well with silt soil slope by denoting that the slope is considered stable. For the sake of argument, the results of the simulation appear to correlate very well with the observation in sunny conditions, however, further supporting simulation in rainy conditions are documented below.

#### Simulations of the FoS Using SLIDE Model under Rainy Conditions in Clay Loam Soil Slope

The last simulations were undertaken to verify the performance of the typical clay loam soil. The results of the simulation, in this case, also suggest an unstable slope under heavy raining conditions. Low FoS values were estimated throughout using different methods as illustrated in Figures 27–30. This means that clay loam slopes in the Thulamela area are prone to failure when exposed to typical local heavy rainfalls.

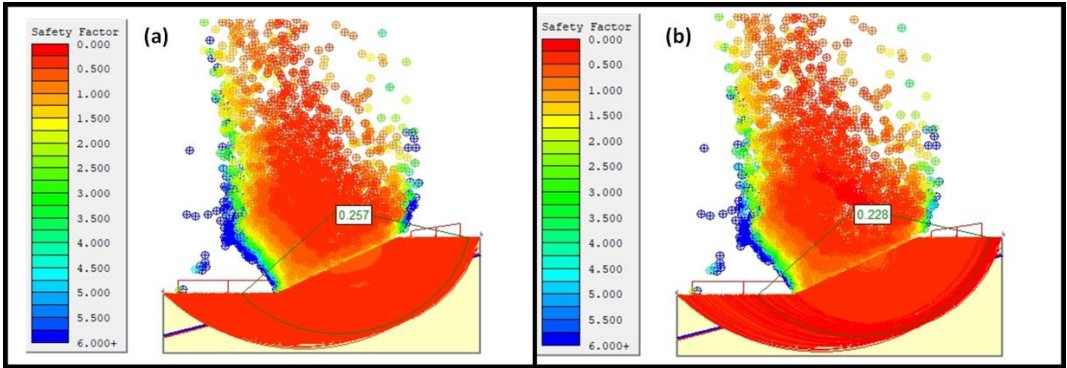

**Figure 27.** Safety factor simulated for drained to undrained conditions in clay loam soil slope using (**a**) Bishop's simplified method and (**b**) Janbu's simplified method.

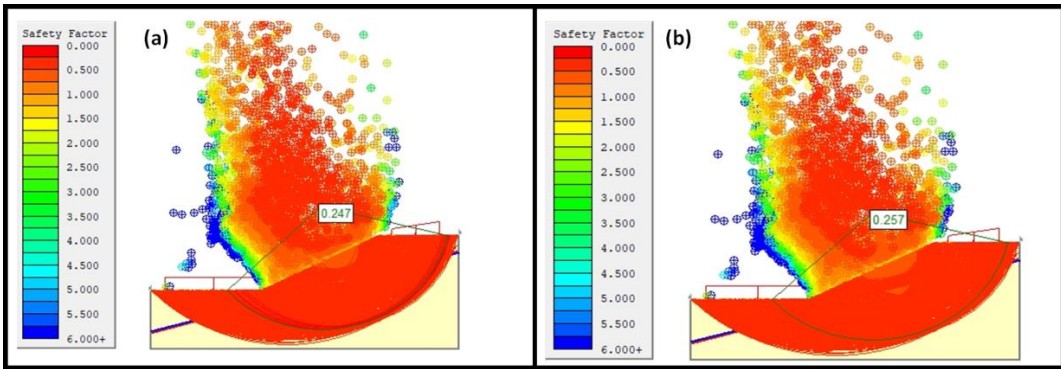

**Figure 28.** Safety factor simulated for drained to undrained conditions in clay loam soil slope using (**a**) Spencer's method and (**b**) Corp of Engineers' Number One method.

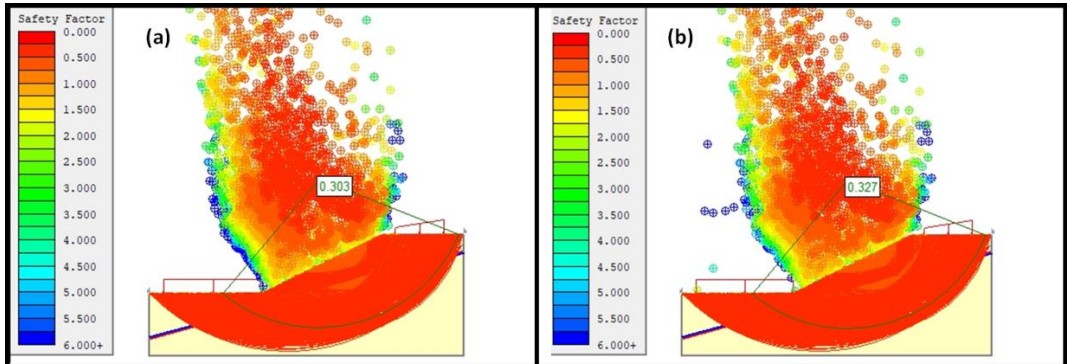

**Figure 29.** Safety factor simulated for drained to undrained conditions in clay loam soil slope using (**a**) Corp of Engineers' Number Two method and (**b**) the Lower Karafiath method.

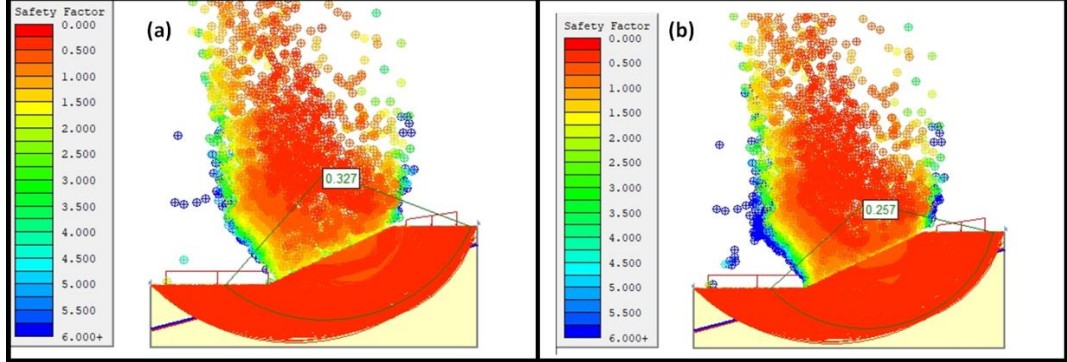

**Figure 30.** Safety factor simulated for drained to undrained conditions in clay loam soil slope using (**a**) the Lower Karafiath method and (**b**) the Gle/Morgenstern Price method.

Simulations of the FoS Using FLACSlope (FDM) Model under Sunny to Rainy Conditions in Clay Loam Soil Slope

Results show that slope was stable in sunny conditions (see Figures 31–34 and Table 7), meanwhile the stability of the slope gradually reduced with the increase in the water table. The simulation in Figure 33 suggests that the selected slope is prone to failure provided it is exposed to extreme rainfalls. It is arguable that the gradual drop on FoS is relative to the increase in pore water pressure, and as already indicated, the failure observed in the study area has been noted to occur when soil phases changes due to extreme rainfalls. Therefore, one can draw an overall conclusion that the extreme rainfall is indeed one of the common triggering factors of slope instability.

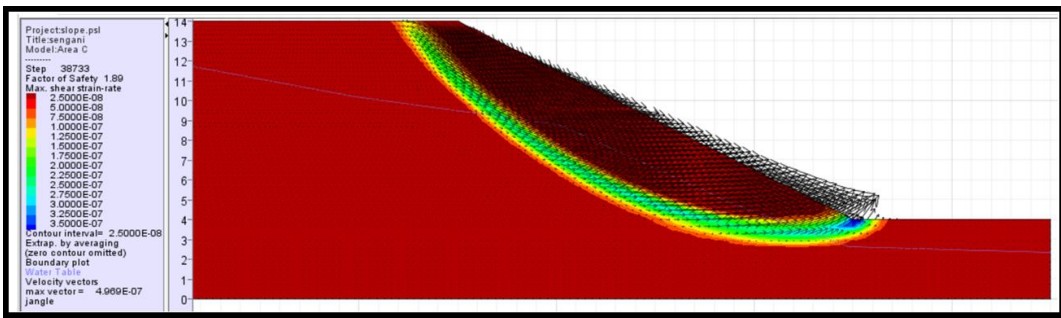

**Figure 31.** Simulated FoS of the clay loam soil slope using FLACSlope in sunny with a water table with 1000 m$^3$ volume of water. Note that maximum shear strain simulated ranges between $2.500 \times 10^{-8}$ to $4.989 \times 10^{-7}$.

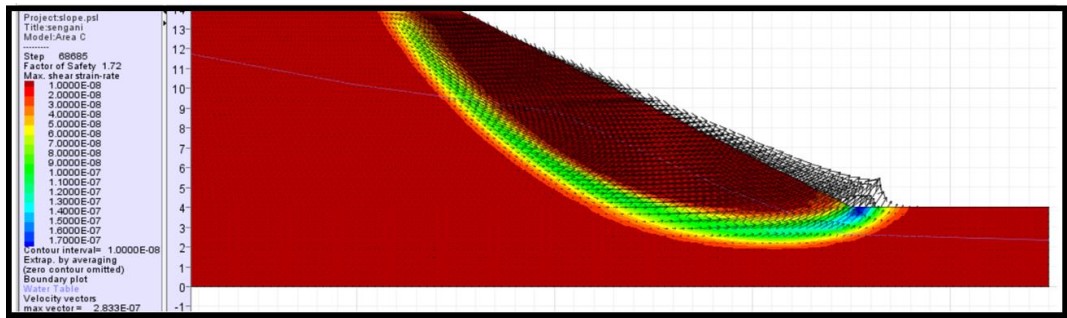

**Figure 32.** Simulated FoS of the clay loam soil slope using FLACSlope in rainy conditions with a water table with 1500 m$^3$ volume of water. Note that maximum shear strain simulated ranges between $1.0 \times 10^{-8}$ to $2.833 \times 10^{-7}$.

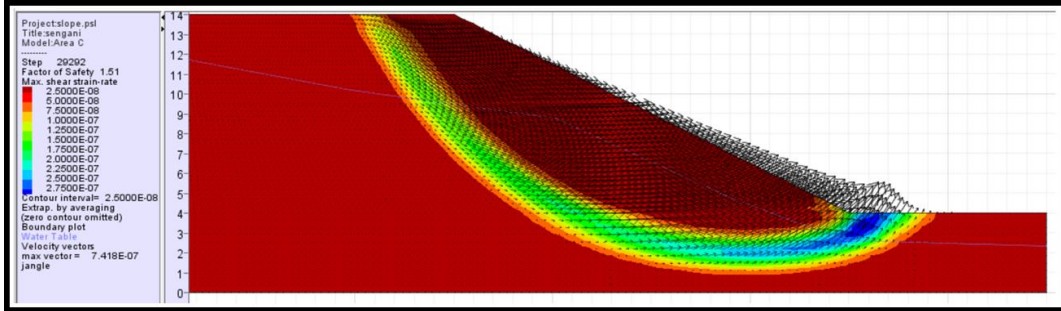

**Figure 33.** Simulated FoS of the clay loam soil slope using FLACSlope in rainy conditions with a water table with 2000 m$^3$ volume of water. Note that maximum shear strain simulated ranges between $2.500 \times 10^{-8}$ to $7.418 \times 10^{-7}$.

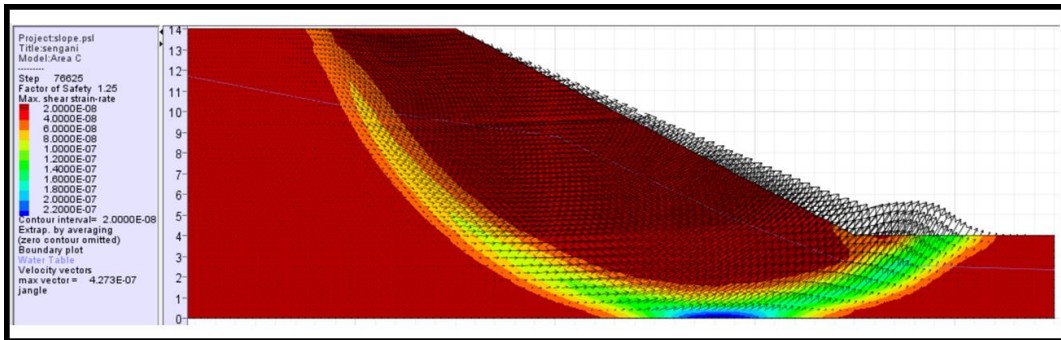

**Figure 34.** Simulated FoS of the clay loam soil slope using FLACSlope in rainy conditions with a water table with 2500 m$^3$ volume of water. Note that maximum shear strain simulated ranges between $2.00 \times 10^{-8}$ to $2.273 \times 10^{-7}$.

**Table 7.** General results of FLACSlope simulations in clay soil slope.

| Factor of Safety (FoS) | Volume of Rainfall (m³) | Slope Max Shear Strain (Displacement in Shear) | |
| --- | --- | --- | --- |
| | | Minimum Shear Strain | Maximum Shear Strain |
| 1.89 | 1000 | $2.500 \times 10^{-08}$ | $4.989 \times 10^{-07}$ |
| 1.72 | 1500 | $1.0 \times 10^{-08}$ | $2.833 \times 10^{-07}$ |
| 1.51 | 2000 | $2.500 \times 10^{-08}$ | $7.418 \times 10^{-07}$ |
| 1.25 | 2500 | $2.00 \times 10^{-08}$ | $2.273 \times 10^{-07}$ |

Based on observations, it was noted that post landslides create land fractures as long as 10 m. The uncontrolled surface drainage water gets channeled toward these fractures resulting in more unstable slopes. Although some slopes have been supported, the support system has been observed to fail dramatically. Perhaps the most important point to make is that further research is needed to probe the ineffective support systems.

4.5.4. Simulation of Pore Pressure Variation Analysis in Case of Soil Slope

Exploring the effect of pore water pressure on the stability of the soil slope is one of the critical aspects to assess. Nevertheless, numerical simulation on variation in pore water pressure relative to slope stability with a change in the water table has been simulated. Considering that the previous section focused more on FoS, it is equally important to have a close look at the variation of pore water pressure with change in the water table, as this factor also contributes largely to the generated FoS.

With reference to Figures 35–38, the simulation has denoted that pore water pressure contour plots were gradual increasing with the increase with water table while the FoS decreases on the other hand. This brings us to talk about the effect of pore water pressure on stress interacting among soil particles as well as reducing the connection between soil particles. In short, the presents and increase in pore water in the pores (or voids) of soil, the pore water pressure generally pushes the soil particles apart and reduces the stress between the particles resulting in soil slope failure. It is well-known that pore water pressure reduces engineering material (soils) strength by reducing their cohesion and frictional resistance. Al-Karni [40] has evidence the above-mentioned behavior of water pore pressure toward engineering material, the previous documented that the maximum deviator stress decreases as the pore water pressure increases.

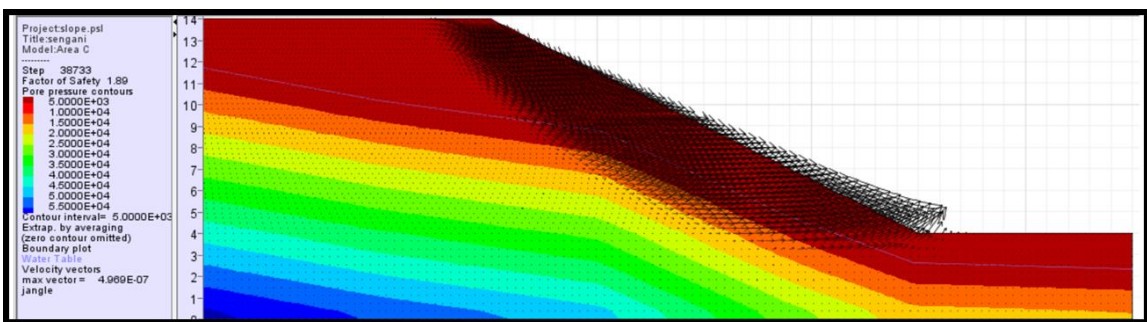

**Figure 35.** Simulated water pore pressure of the soil slope using FLACSlope in sunny conditions with a water table with 1000 m³ volume of water.

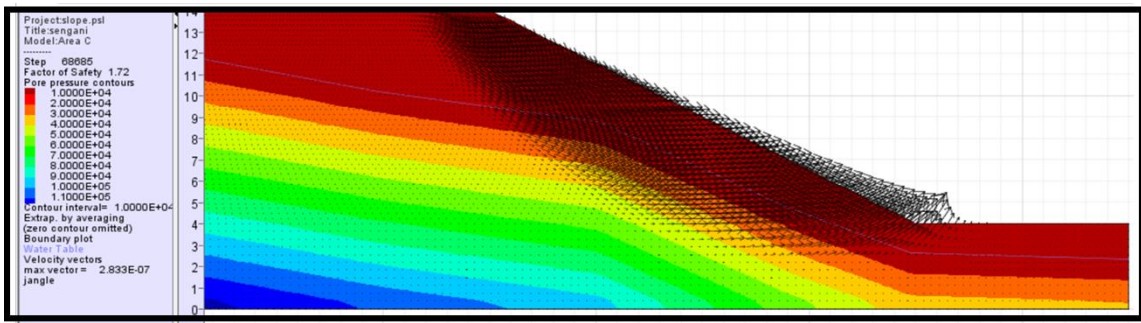

**Figure 36.** Simulated water pore pressure of the soil slope using FLACSlope in rainy conditions with a water table with 1500 m$^3$ volume of water.

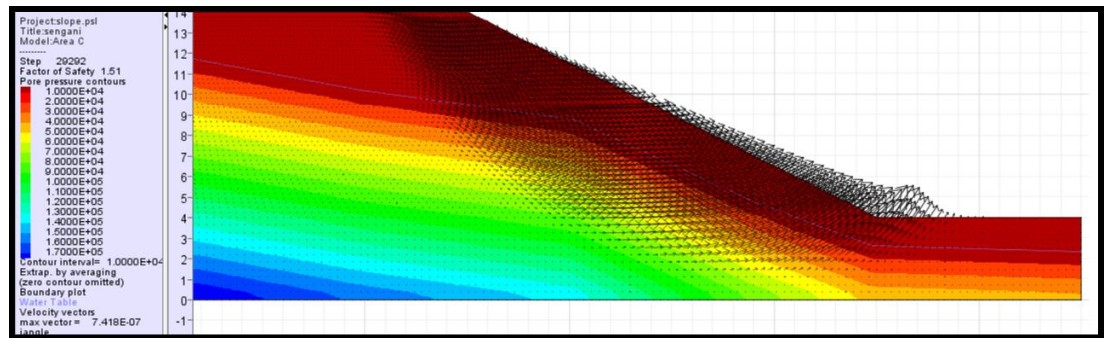

**Figure 37.** Simulated water pore pressure of the soil slope using FLACSlope in rainy conditions with a water table with 2000 m$^3$ volume of water.

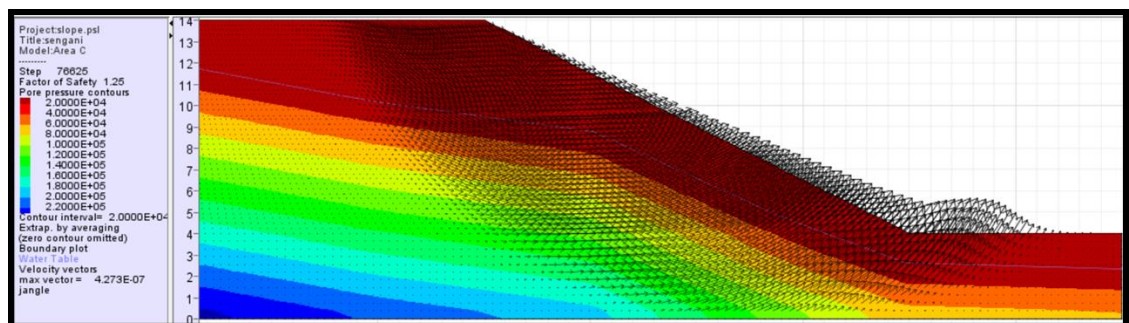

**Figure 38.** Simulated water pore pressure of the soil slope using FLACSlope in rainy conditions with a water table with 2500 m$^3$ volume of water.

It is therefore arguable that the entire simulations agree with the visual and laboratory analysis, most especially the factor that low FoS factors with extreme displacement have been simulated with an increase in the water table or during extreme rainfalls. This correlation can therefore validate and solidify the confusion which has been not well understood within the study area. It is also crucial to indicate that extreme rainfalls are not the only factors that influence the soil slope instability in the study area, but the scope of this study is only limited to extreme rainfalls.

## 5. Significance of the Simulation Findings

Several scholars have examined the application of physical or empirical models on the impact of rainfall on slope stability (e.g., [41–56]). Most have argued that the effects of meteorological, morphological, and geological characteristics have been mixed. Indeed, different geographical locations have yielded different results concerning rainfall thresholds for a landslide.

In terms of the results reported in this study, they follow the same pattern as those of past work. For example, Lazzari et al. [11] were able to demonstrate the impct of extreme rainfall on the recurrence

and occurrence of landslides. This is especially true in steep and clay soil areas, as has also been interestingly found in the present study. It may therefore be said that rainfall and other yet unknown factors have much influence on the occurrence of slope instability within the Thohoyandou areas.

The findings reported in this paper provide a broad view of the common mechanisms associated with the recurrence of slope instability in the Thulamela area. In line with the aforesaid, several scholars have conceded that extreme rainfall, active tectonics, and weathering strongly influence rock slope instability [57–78]. This study is no different, with results that compare with previous studies conducted in other areas.

On another note, numerical simulation has been resorted to in this paper is an attempt to understand the overall behavior of slopes. Nevertheless, it is crucial to indicate that numerical software does not replace critical thinking and visual observation. That is why actual data collected from a controlled environment, such as a laboratory or in the field, are required for validation. It is anticipated that the simulation findings presented in this study may guide the compilation of meaningful data for such a purpose.

## 6. Conclusions

The findings of this paper suggest that extreme rainfall that usually occurs between December and January is a triggering factor for landslides along the R518 road in Thohoyandou. Such rainfall can compromise the stability of the road slopes and worsen the high landslide-prone area. Besides the post landslide reported, the area has been observed to slope instability in summer month which also correlate with extreme rainfall within the area during this period. The material behavior changes from that of a solid phase to a liquid one during heavy rainfall. Soil particles lose their bond under extreme rainfall and give way to landslides.

The core of the work was the simulation of slopes of different soil types that have provided results in support of physical observation. Three soil types were simulated: silt clay, clay, and clay loam slopes. Using FEM (SLIDEs) in sunny conditions the slope was simulated to be stable in both silt clay and clay loam, while in clay slope the FoS predicted that the slope is generally unstable or less stable. Further simulations in rainy conditions have shown that in all three cases, FoS values were estimated to be 0.200 to 0.363 signifying a high likelihood for slope failure. Owing to that, the FDM (FLACSlope) model was used as a complementary model, and the simulations have indicated that slopes were stable in sunny conditions. Meanwhile, the stability of the slopes was gradually reduced in rainy conditions and later became unstable or total collapses.

The above simulation using FLACSlope on FoS also incorporated maximum strain taking place across the slope, it was also observed that the displacement was a gradual increase with the increase in the water table. This leads the study to further the simulations by creating pore pressure analysis. Similar to other studies, it was denoted that the water pore pressure increases with the increase in the water table, which also reduces the stress interacting across soil particles, allowing the particles to move freely by reducing the friction among the soil particles. Therefore, it is concluded that the slopes along the R518 road are of high risk when exposed to heavy rainfall. However, mitigation can be implemented to stabilize slopes, including channeling natural water streams around the area. Road walls may be the other mitigating solution, but further research is needed to ascertain their efficacy. It is crucial to indicate that extreme rainfall is not the only contributing slope instability factor in the study area, but the scope of the study is limit to the effect of extreme rainfall on the soil slope.

Indeed, the numerical models used have shown some great correlation among them. However, the model has proven that they can be used for hazard identification of certain slopes by understanding the properties of material climate change within the study area.

**Author Contributions:** Conceptualization, F.S. and F.M.; methodology, F.S. and F.M.; software, F.S.; validation, F.M.; formal analysis, F.S. and F.M.; investigation, F.S. and F.M.; resources, F.S. and F.M.; data curation, F.S. and F.M.; writing—original draft preparation, F.S. and F.M.; writing—review and editing, F.S. and F.M.; visualization,

F.S. and F.M.; supervision, F.M.; project administration, F.S.; funding acquisition, F.S. All authors have read and agreed to the published version of the manuscript.

**Funding:** The two Universities (the University of Limpopo and the University of South Africa) provided funds for this study.

**Acknowledgments:** The authors would like to appreciate the sponsorship given by the two Universities (the University of Limpopo and the University of South Africa). The authors would like to appreciate the assistance and efforts made by Nndanduleni Muavhi in generating the geological map of the study area.

**Conflicts of Interest:** The author wishes to confirm that there are no known conflict of interest associated with this publication, furthermore, there has been no financial support given to influence the outcome of this work.

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
