# Peer review of "Influence of Rainfall Intensity on the Stability of Unsaturated Soil Slope: Case Study of R523 Road in Thulamela Municipality, Limpopo Province, South Africa"

_applsci, doi:10.3390/app10248824_

Round 1
Reviewer 1 Report
Detailed comments:
L44-47 I The discussion about predisposing conditions is only concentrating on studies from Italy. I suggest adding more diverse references that highlighted the relevance of clay minerals and other factors in the past (e.g. Kluger et al. 2017, https://doi.org/10.1130/G38560.1; Chau et al. 2004, https://doi.org/10.1016/j.cageo.2003.08.013).
L55 Please add the country in which your study was performed to the description of your study area.
L127 Please add the standards you were following to obtain grain size distribution curves and Atterberg limits
Figures 3, 4, 5, 6, 7 Photographs are of low resolution. Please improve visibility of all photographs
Figure 10 The grain size distribution curves are of low resolution. The labels as displayed in the legend are not differentiable.
L327 The authors may have misinterpreted their grain size distribution curves. As I see it, the samples displayed in figure 10 have relatively low amount of fines content as the fraction below 1mm is between 20 - 30%. Most of the material is coarser than 2mm making it coarse sand. They are definitely not ‘mostly consisting of clay material’. When considering the Atterberg limits, I would question the validity of grain size analyses performed in this study, because sandy soils commonly would have very low plasticity indices.
L385 As I understood it, the authors performed additional laboratory tests (triaxial tests, oedometer tests) to derive mechanical properties for their numerical models. However, they did not provide experimental evidence of the results presented in Tab. 3. So, the reader has to simply believe that the experiments were performed correctly. I suggest that the authors present all their experimental results or that they refer to a publication in which one can find those experimental results.
L411 It was not terribly clear to me how the authors linked rainfall events to landslide triggering. I suggest the authors follow established approaches, such as the ones summarized in Segoni et al. 2018; Landslides, in order to find rainfall thresholds relevant in their regional setting. The effect of unsaturated soil behaviour was not considered (although it was mentioned in the title of the study).
L466 The section about simulation results should be enhanced. I suggest to better link the simulation results to the scenarios summarized in Tab. 4. I further suggest that the authors find a way how to visualize their results in a comparative diagram. This would make it easier for the reader to follow and compare the results made.
Figs. 16 – 25 all these figures need to be of higher resolution. I suggest to reduce the number of figures and shift some of them to the appendix or online repository.
Reviewer 2 Report
The paper should be substantially reduced to target on its topic, i.e., the influence of rainfall intensive to slope failure. The selected case studies involve too many unnecessary details, with many mixed content and simulation techniques. The reviewer found difficult to get the point of this paper as a research article. Please prepare for a more condensed version for reconsideration.
Reviewer 3 Report
The study entitled “Influence of rainfall intensity on the stability of unsaturated soil slope.” This work overall have some interest to the scientific community and deserves to be published, but there are some severe adjustments that need to be done before publication. For this purpose, some significant comments are listed below.
- Line 15. Use factor of safety to replace FoS.
- There are 25 figures in the manuscript. Too many figures may be difficult for readers and made the manuscript looks fragmented. I suggest to reduce the figures number down to twenty.
- Line 52. “However, the mechanism associated with the recurrence of these slope instability events is not well defined.” The sentences didn’t clearly define the knowledge gap on .
- Line 416. Please define the “rainfall departure.” Does the rainfall departure used to simulate FoS?
- Line 483. Please provide more notation and figure caption on Figure 16 and Figure 17, especially on the dot-plot.
- The legend in Figure 18, 21, and 24 is unclear.
- To highlight the interest to the scientific community in the world, I suggest authors should refer and compare other literatures on the triggering and predisposing factors of landsliding. I listed some related literatures as below:
- Imaizumi et al. (2015) Temporal and spatial variation of infilling processes in a landslide scar in a steep mountainous region, Japan. Earth Surface Processes and Landforms, 40(5), 642-653.
- Goswami et al. (2011) Distribution and causes of landslides in the eastern Peloritani of NE Sicily and western Aspromonte of SW Calabria, Italy. Geomorphology, 132(3-4), 111-122.
- Bucci et al. (2016) Landslide distribution and size in response to Quaternary fault activity: the Peloritani Range, NE Sicily, Italy. Earth Surface Processes and Landforms, 41(5), 711-720.
- Chen et al. (2019) Controls of preferential orientation of earthquake- and rainfall-triggered landslides in Taiwan's orogenic mountain belt. Earth Surface Processes and Landforms, 44(9), 1661-1674.
- Guo et al. (2015) Quantitative assessment of landslide susceptibility along the Xianshuihe fault zone, Tibetan Plateau, China. Geomorphology, 248, 93-110.
- Line 730. “Mitigation can be implemented to stabilize slopes including channeling natural water streams.” It can be supported by simulation results?
- In the conclusions, author could provide a critical rainfall of stability based on the simulation results which is useful and can be applicable for the hazard mitigation.
Reviewer 4 Report
The manuscript entitled "Influence of rainfall intensity on the stability of unsaturated soil slope" presents a slope stability case study in South Africa. It can be recognized that the work of the study took quite a bit of time. However, I provide my comments below, which reveal the problem of the paper.
- The title looks like a general study but the whole manuscript is a case study. And the conclusion is specific to the study area. Therefore, I suggest that the title should be modified.
- The Introduction part is poor. The review of previous studies is not organized and presented well. The authors should improve it.
- Line 47-49: The authors mentioned, “Nonetheless, understanding the impact of extreme rainfall on slope stability in an unsaturated soil is still to be improved.” However, I disagree with this statement. In my opinion, the researches in the field are abundant. And It conflicts with Line 696-698. I suggest the author read the following paper and find another way to highlight the importance of the work. Bogaard, T. A.; Greco, R. Landslide Hydrology: From Hydrology to Pore Pressure. WIREs Water 2016, 3 (3), 439–459. https://doi.org/10.1002/wat2.1126.
- In the last paragraph of the introduction, usually, the structure of the paper should be introduced. What is the methodology? What is the tool? It does not follow the scientific English writing structure.
- Please add a flow chart and descriptions of the whole study. The study is combined with several works but hard to quickly understand the flow of the study.
- Legends and texts of lots of Figures are not clear. Please improve the resolution of figures or modify the size of the texts.
Round 2
Reviewer 1 Report
Please perform a careful english check on the sentences you added in your revised manuscript. I identified a large number of spelling mistakes and grammar errors. Some examples are listed below:
"Bogaard and Greco [79] have listed rainfall as one of the most common landslide widespread triggering hazards in the world. The previous author supported their arguments [...]"
In first sentence, syntax is not correct. In the second sentence to what are "the previous authors" and "their" referring to?
"Furthermore, previous authors also documented that the “extraordinary precipitation events trigger most of the landslides, but, at the same time, the vast majority of slopes do not fail” the previous suggestion correlates very well some studies such as those of Sengani and Mulenga [76]; Sengani and Zvarivadza [64] and Mutanamba [47]."
Not clear to whom "previous authors" refer to. Syntax not correct here.
"Sophisticated methods such as numerical simulated has been implemented to understand sliding of clay soil material Kluger et al. [80], and revealed that “the high sensitivity and contributes to an improved understanding of the mechanisms of flow sliding in sensitive, altered tephras rich in spheroidal halloysite”."
Syntax not correct throughout the sentence and many spelling mistakes were made here.
Author Response
Kindly find attached replies.
Thanks

Reviewer 3 Report
The manuscript has been largely improved. I suggest to accept for publication in present form.
Author Response
Dear Reviewer
We appreciated your effort and constructive suggestions.
Thanks

Reviewer 4 Report
- In my opinion, the manuscript was not revised well. I agree with the comment of another reviewer that the manuscript should be substantially reduced. I do not appreciate the manuscript because of the present and quality of the paper. 10 thousand words and 38 Figures are too much for a paper.
- In ch3, too many details are presented. Some are very elementary. I do not understand why the tutorial of sieve analysis should be introduced.
- Several figures have the same caption. For example, Fig 35-38, 31-34, 22-26. There's not enough information on the Figures. It appears that the manuscript is a very rough treatment.
- I cannot read and understand the Figures of FLACSlope results. What makes the results of Figure 15- Figure18 different? Fig. 22-26? What are the different scenarios? It is totally unclear.
- Parts of the conclusion, cannot be identified by the models. For example, Line 888-889: “The material behavior changes from that of a solid phase to a liquid one during heavy rainfall.” I am doubtful about how the authors get the idea.
- The authors used both FEM and FDM and pointed out the differences in the results. However, if the paper wants to compare the different methods, not only the difference in results should be pointed out but the reason, mechanisms, schemes, theories of the numerical methods should be compared and explained.
- The author should further consider how to present the research results.
Author Response
Kindly find attached replies.
Thanks
